# Contraction and Hourglass Persistence for Learning on Graphs, Simplices, and Cells

**Mattie Ji**[1,*], **Indradyumna Roy**[2,3,*], **Vikas Garg**[2,4]
[1]University of Pennsylvania. [2]Aalto University. [3]Indian Institute of Technology Bombay. [4]YaiYai Ltd.
`mji13@sas.upenn.edu, indradyumna.roy@aalto.fi, vgarg@csail.mit.edu`

## Abstract

Persistent homology (PH) encodes global information, such as cycles, and is thus increasingly integrated into graph neural networks (GNNs). PH methods in GNNs typically traverse an increasing sequence of subgraphs. In this work, we first expose limitations of this inclusion procedure. To remedy these shortcomings, we analyze contractions as a principled topological operation, in particular, for graph representation learning. We study the persistence of contraction sequences, which we call Contraction Homology (CH). We establish that forward PH and CH differ in expressivity. We then introduce Hourglass Persistence, a class of topological descriptors that interleave a sequence of inclusions and contractions to boost expressivity, learnability, and stability. We also study related families parametrized by two paradigms. We also discuss how our framework extends to simplicial and cellular networks. We further design efficient algorithms that are pluggable into end-to-end differentiable GNN pipelines, enabling consistent empirical improvements over many PH methods across standard real-world graph datasets. Code is available at https://github.com/Aalto-QuML/Hourglass.

## 1 Introduction

Graph Neural Networks (GNNs) (Hamilton, 2020) is a powerful paradigm for learning on structured data, yet their expressive power is fundamentally limited by the Weisfeiler–Lehman (WL) hierarchy (Xu et al., 2019; Morris et al., 2019). In particular, message-passing GNNs struggle to capture higher-order topological signals (ex. the presence, interaction, and disappearance of cycles) that often drive downstream performance in molecular learning, physical systems, and network science (Garg et al., 2020; Chen et al., 2020; Tahmasebi et al., 2021). Persistent Homology (PH) (Edelsbrunner et al., 2002) from Topological Data Analysis (TDA), offers a way to extract such signals by tracking the life of topological features in a filtration. Accordingly, PH-based descriptors are increasingly used to augment GNNs and higher-order neural architectures (Papamarkou et al., 2024), and to boost expressivity (Ballester & Rieck, 2024; Zhang et al., 2025; Li & Leskovec, 2022).

However, many PH pipelines in graph learning rely on *inclusion-based filtrations* ("forward persistence") (Rieck et al., 2019; Hofer et al., 2020; Immonen et al., 2023; Ballester & Rieck, 2024; Ying et al., 2024; Ji et al., 2025a;b). This one-sided view can leave information on the table. For instance, a forward pass on graphs, *cycles born can never die*; and while *new connected components can appear* as vertices arrive. This is also undesirable from the perspective of *metrizability*, as bottleneck distances can be ill-defined due to the presence of different number of permanent features.

To resolve this, we draw our insights from topological *contractions*. Contractions have a wide variety of uses in the study of persistent homology. Discrete Morse theory (Forman, 1998; 2002) uses the idea of simple contractions/collapses to simplify complexes. Contractions have since been used in many contexts to speed up PH calculations (Dłotko & Wagner, 2013; Botnan & Spreemann, 2015; Boissonnat & S., 2016; Dey et al., 2016; Dey & Slechta, 2020; Boissonnat et al., 2023). The PH literature also has ways to resolve the aforementioned metrizability problem, one of which being extended persistence (Agarwal et al., 2004; Cohen-Steiner et al., 2009; Bermingham et al., 2023; Turner et al., 2024) by considering both a forward and reverse pass on the input data.

---

*Equal Contributions.

| **Persistence Goes Forward and Backward (Section 3):** | |
| --- | --- |
| Construction of Backward PH and Forward-Backward PH | Definition 6, Definition 7 |
| Incomparability for Forward vs. Backward PH | Proposition 1 |
| Construction of $(\sigma, \tau)$-FB PH and Hourglass Persistence | Definition 8, Definition 9 |
| Hourglass PH $\succ$ FB-Persistence $\succ$ Forward + Backward PH | Theorem 1, Proposition 2 |
| **Relation to Extended Persistence and Extensions (Section 4):** | |
| Introduce $(f, g)$-FB Persistence, with $(\sigma, \tau)$-FB as Example | Definition 10, Proposition 4 |
| FB-Persistence $\neq$ Extended Persistence | Proposition 5 |
| $(f, f)$-FB persistence $=$ Forward PH in $f$ | Proposition 6 |
| **Extensions to Higher Complexes and Stability (Section 5, 6):** | Proposition 7; Theorem 2 |
| **Algorithmic Design and Experiments (Section 7):** | Alg 1, Alg 2; Table 1, 2 |

Figure 1: Overview of the paper. Each row summarizes a core item and the section where it appears.

In this work, we study the PH of sequences of contractions (which we refer to as *contraction homology* (CH)) for graph representation learning and their higher-order counterparts. While PH can be thought of as a *time-forwarding process*, CH can be thought of as a *time-reversing process* that operates by *contracting* substructures. We then study a particular form of CH which we call *backward persistence* (BH), defined by contracting a sequence of *intermediate complexes*. We compare the expressivity of forward PH and backward PH and show that neither subsumes the other.

Motivated by our discovery, we then propose *Forward–Backward (FB) persistence*, which *concatenates* an inclusion phase with a contraction phase. Intuitively, FB links *how features appear* (forward) with *how they subsequently vanish* (backward), assigning finite lifetimes to structures that would otherwise persist to infinity in forward PH. We show FB-persistence is *strictly more expressive* than forward and backward considered in isolation.

Because of the modular nature of how we set up the intermediate complexes, we can interchange the sequence of intermediate complexes that are included and contracted at each step. Guided by this combinatorial insight, we introduce *Hourglass persistence*, which *interleaves* inclusions and contractions in *arbitrary order*, subject only to the constraint that a piece must be included before it can be contracted. Hourglass persistence also gives a tradeoff between runtime and PH information of the whole complex (see Section E.4), which may be beneficial for evaluating large graphs.

Finally, we propose a general framework of $(f, g)$-FB persistence with respect to two filtration functions $f$ and $g$, which extends both extended and FB-persistence. We then extend our framework to higher-order structures, design algorithms to compute $(f, g)$-FB persistence over graphs, and integrate them into GNNs for empirical validations.

Our work differs from many prior works in several ways. (1) The typical usage of collapses to speed up PH calculations is for a sequence of inclusions and aims to alter the output as minimally as possible. Instead, we study the PH of contractions that can significantly alter the topology and their interactions with inclusions. (2) Rather than contracting superlevel sets in BH, we make the substructures more granular by quotienting *intermediate complexes*. In fact, we show in Proposition 5 that FB-persistence and extended persistence are different. (3) To the best of our knowledge, extended persistence used in GNNs (Zhao & Wang, 2019; Zhao et al., 2020; Carriere et al., 2020; Yan et al., 2022) does not compute their descriptors using the contraction perspective for experiments, but we will demonstrate the utility of contractions in the experiments.

We summarize our paper in Figure 1 and leave proofs and additional details to the Appendix.

## 2 BACKGROUND AND PRELIMINARIES

Let $G = (V, E)$ be a finite undirected graph, possibly with self-loops and multi-edges. A common theme of enhancing graph neural networks (GNNs) is to incorporate persistent topological descriptors derived from a given filtration of $G$. For our purpose, a filtration is as follows.

**Definition 1.** *Let $f : V \cup E \to \mathbb{R}$ be a function. We say $f$ is a **filtration function** on $G$ if for every edge $e = (v, w)$, $f(v) \leq f(e)$ and $f(w) \leq f(e)$. The ordered set $\{f^{-1}((-\infty, t])\}_{t \in \mathbb{R}}$ has only finitely many elements, which gives a list of sequential subgraphs:*

$$G_{-1} = \emptyset \subset G_0 \subset G_1 \subset ... \subset G_n = G.$$

*We say $f$ is a **vertex-based filtration** if for every edge $e = (v, w)$, $f(e) = \max(v, w)$. We say that $f$ is an **edge-based filtration** if $f(v) = 0$ for all $v \in V$ and $f(e) > 0$ for all $e \in E$.*

For consistency, the function $f$ should not change after relabeling the vertices and edges, rather $f$ should only depend on the intrinsic features the graph $G$ comes with. This property is called **permutation equivariance** (Ballester & Rieck, 2024). Common examples includes degree filtration.

**Definition 2.** *Suppose $G$ is a **colored graph** (i.e., vertices have labels) equipped with an additional structure of a coloring function $c : V \to C$, where $C$ is a collection of colors. A **vertex-color filtration** is a vertex-based filtration $f : V \cup E \to \mathbb{R}$ such that $f(v) = f(w)$ for all $v, w$ with $c(v) = c(w)$. Intuitively, $f$ preserves the vertex coloring.*

A special case of Example 2 includes degree filtration where $c$ is the degree function and $f = c$. A common scenario where colored graphs arise is from molecular graphs (Hoogeboom et al., 2022; Xu et al., 2022; 2023; Song et al., 2024), where the colors are the type of atoms (e.g., oxygen, carbon).

A common descriptor to extract data from a given filtration is **persistent homology**.

**Definition 3.** *Let $X_\bullet = (X_0 \xrightarrow{f_1} X_1 \xrightarrow{f_2} ... \xrightarrow{f_{n-2}} X_{n-1} \xrightarrow{f_n} X_n)$ be a sequence of topological spaces (ex. **graphs**) $X_i$ with maps $f_i : X_i \to X_{i+1}$. The **$k$-th persistent homology** of $X_\bullet$ is the result of applying $H_k(-)$ (in $\mathbb{Z}/2$-coefficients) to $X_\bullet$, that is, it is the sequence of linear maps:*

$$H_k(X_\bullet) = H_k(X_0) \xrightarrow{(f_0)_*} H_k(X_1) \xrightarrow{(f_1)_*} ... H_k(X_{n-1}) \xrightarrow{(f_{n-1})_*} H_k(X_n).$$

*An element $v \in H_k(X_i)$ is said to **born** at $i$ if $v \notin \mathrm{im}((f_{i-1})_*)$. We say $v$ **dies** at $d \geq i$ if $d$ is the first element such that $(f_{d-1})_* \circ ... \circ (f_i)_*(v) = 0$. The **persistence pair** associated to $v$ is the pair $(b, d)$. If no such $d$ exists, we mark the pair as $(b, \infty)$. If the incoming complex is of the form $X_\bullet = (X_{-1} = \emptyset \xrightarrow{f_{-1}} X_0 \to ... \to X_n)$ (e.g. from a filtration), by $H_k(X_\bullet)$ we mean $H_k(-)$ of the sequence with the $X_{-1}$-term truncated.*

*Suppose the vector spaces are all finite dimensional, the $k$-th **persistence diagram** (i.e., barcodes in Ghrist (2007)) of $X_\bullet$ is the collection of persistent pairs for a specific choice of basis of the $H_k(X_i)$'s obtained from an interval decomposition of $H_k(X_\bullet)$ (see Theorem 2.7-8 of Chazal et al. (2016) or Theorem 4.7 of Lesnick (2025) for more details).*

Appendix B.1 shows its equivalence to the usual definition of graph PH. Here $H_0(-)$ are connected components, $H_1(-)$ are independent cycles, $H_2(-)$ are voids, and $H_k(-)$ are $k$-dimensional voids.

**Remark 1.** *Explicitly, the $k$-th persistence diagram of $X_\bullet$ can be computed by first picking a non-zero vector $v$ of $H_k(X_0)$, and consider the sequence of linear subspace generated by the iterated image of $v$ in $H_k(X_\bullet)$ until it becomes $0$. Then we remove this sequence of linear subspaces from $H_k(X_\bullet)$. If there is still a non-zero vector $w$ in $H_k(X_0)$, we pick $w$ and repeat the process (if the linear map sends $w$ to the complement, we consider it becomes $0$). Otherwise, we choose a non-zero vector from what is left in $H_k(X_1)$ (if it exists) and look at its iterated images ahead, and so on.*

## 3 PERSISTENCE GOES FORWARD AND BACKWARD

For now, we will focus on the case of graphs, but many constructions here can be readily generalized to the setting of simplicial and cell complexes (see Section 5). Classically, there are numerous applications in TDA and GNNs that uses an inclusion-based persistent homology (Edelsbrunner & Harer, 2008; Edelsbrunner & Morozov, 2012; Immonen et al., 2023; Ballester & Rieck, 2024). Concretely, an inclusion-based PH may be viewed as an example of Definition 3 as follows.

**Example 1** (Inclusion-based/Forward PH). *Let $f : G \to \mathbb{R}$ be a filtration function, and consider applying $H_0(-)$ and $H_1(-)$ to $G_\bullet : G_{-1} = \emptyset \subset G_0 \subset ... \subset G_n = G$, where each map is the inclusion map. Then the **persistent diagram** in Definition 3 recovers all birth/death pairs $(b, d)$ such that $b < d$ (i.e., it omits trivial births). See Appendix B.1 for a formal proof. We can recover the trivial births by counting the number of unmarked simplices at each time.*

The interest that initiated this work was: what if we reverse the inclusion steps with **contractions**?

**Definition 4.** *Let $G$ be a graph, the **contraction homology** (CH) of $G$ is the persistence diagrams associated to any sequence of subgraph contractions of $G$ down to a single point.*

Figure 2 illustrates an example of contraction homology. For now, we would like to focus on a specific class of CH that contracts the following sub-graphs known as *intermediate complexes*.

**Definition 5.** *Let $H \subset G$ be a subset. The closure of $H$ is the union of $H$ and all vertices with incident edges contained in $H$. Let $\emptyset = G_{-1} \subset G_0 \subset ... \subset G_n = G$ be a filtration of $G$ induced by $f$, we define the **intermediate complexes** $\mathrm{IC}_i(G, f)$ as the closure of $G_i - G_{i-1}$ for $i = 0, ..., n$. We omit the symbol $f$ when the context is clear.*

**Definition 6** (Backward PH). *Let $G_\bullet : G_{-1} = \emptyset \subset G_0 \subset ... \subset G_n = G$ be a filtration of $G$, we consider a **sequence of contractions** with respect to $G_\bullet$ as $(G_\bullet)^v$ below:*

$$G \to G_{n+1} := G/\mathrm{IC}_n(G) \to G_{n+2} := G/(*_{n+1}\cup\mathrm{IC}_{n-1}(G)) \to G/(*_n\cup\mathrm{IC}_{n-2}(G)) \to ... \to *,$$

*where $*_{n+1}$ is the point representing the total contracted subcomplex in the previous step, and so on. The intermediate maps are the natural quotient maps. We define the $i$-th backward persistent diagram as the persistent diagram associated with $H_i((G_\bullet)^v)$.*

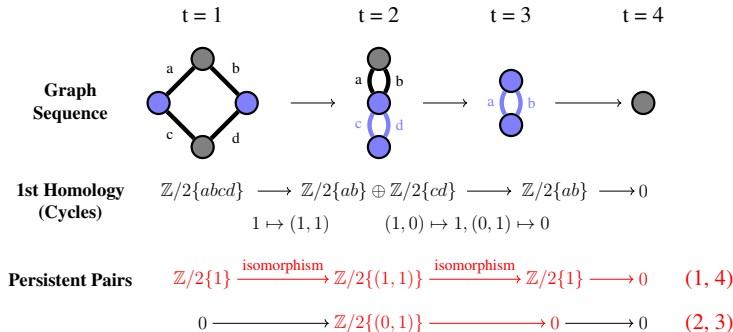

Figure 2: Contraction Homology of a graph $G$. The blue simplices indicate what is contracted.

We observe that the backward PH introduces new information that forward PH may ignore. A reason for this distinction is that a cycle that is born in forward PH can never die, but backward PH can kill cycles. Conversely, backward PH can never make a new connected component apart from the initial step, but forward PH can spawn new components later.

**Proposition 1.** *There exists graphs $G, H$ with permutation equivariant filtrations such that the forward PH of their filtrations cannot tell apart $G, H$ but backward PH can. Similarly, there are graphs that backward PH cannot tell apart but forward PH can. Examples can be found in Figure 3.*

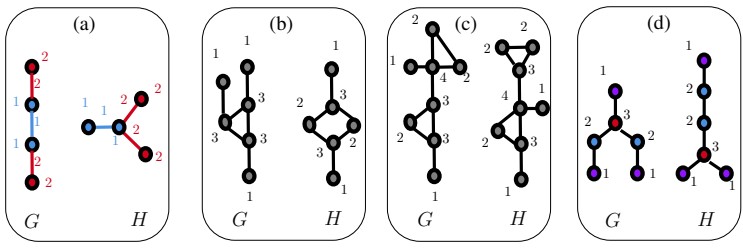

Figure 3: Example graph pairs where (a) color filtration with same **forward-based PH** but different **backward-based PH**, (b) degree filtration with same **backward-based PH** but different **forward-based PH**, (c) degree filtration with same **forward and backward PH** but different **FB-persistence**, (d) degree filtration with same **FB-persistence** but different **hourlgass persistence**.

Another motivation for us to introduce backward-based PH is from the viewpoint of **metrizability** and **stability**. Classically, the **bottleneck distance** Cohen-Steiner et al. (2006) between 2 persistence

diagrams $\text{PD}_1, \text{PD}_2$ of the same size is the infimum of the value $\max_{p \in \text{PD}_1} ||p - \pi(p)||_\infty$ where $\pi$ ranges over all bijections $\pi : \text{PD}_1 \to \text{PD}_2$. The bottleneck distance is finite for persistence diagrams coming from the same graph, but for diagrams, coming from two different graphs, this distance may very well be infinite. The reason why is because the number of tuples of the form $(-, \infty)$ for both graphs may be different, which adds an $\infty$ to the distance function.

One might argue that there can be workarounds by setting the time to die at a finite time $N$ after both filtrations ended, or make it die at $-1$. These modifications however can be quite sensitively altered by, for example, having many features die around $N$ or $-1$. One solution would be to extend the (inclusion-based) PDs by killing off the permanent features topologically via a sequence of contractions. Formally, this motivates our definition of **Forward-Backward (FB) persistence**.

**Definition 7** (Forward-Backward Persistence). *Let $G_\bullet = (\emptyset = G_{-1} \subset G_0 \subset ... \subset G_n = G)$ be a filtration of $G$. The $i$-**th FB-persistence** is the diagram $H_i(G_\bullet + (G_\bullet)^v)$, where $G_\bullet^v$ denotes the sequence of contractions in Definition 6 and $+$ denotes the concatenation of the two sequences.*

We now establish the expressivity benefits due to FB-persistence.

> **Theorem 1.** *FB-persistence is **strictly more expressive** than forward and backward persistence combined. More precisely, for any graph $G$ with filtration $f$, the FB-persistence with respect to $f$ can recover the correspondent forward and backward persistence. However, there exists graphs $G, H$ with a permutation equivariant filtration $f$ such that their forward and backward persistence are the same, but their FB-persistence differ (see Figure 3(c)).*

A high level reason why Theorem 1 holds is that deciding how we concatenate the values from the forward persistent diagrams with the values from the backward persistent diagrams is rather subtle.

Motivated by the perspective of **color-based filtrations** (Ballester & Rieck, 2024; Immonen et al., 2023; Ji et al., 2025a), we observe that there is no canonical reason to contract in the order of $\text{IC}_n(G), \text{IC}_{n-1}(G), ...$ in Definition 6 and Definition 7. Since the intermediate complexes satisfy (i) $\bigcup_{i=0}^n \text{IC}_i(G) = G$ and (ii) $\text{IC}_i(G)$ and $\text{IC}_j(G)$ can only intersect at vertices, any sequence of contracting the subgraphs in the list $\text{IC}_0(G), \text{IC}_1(G), ..., \text{IC}_n(G)$ would have terminated at the single point set $*$. Likewise, we can also see the filtration step as a special case of spawning the pieces $\text{IC}_0(X), \text{IC}_i(X), ...$ in ascending order. There is no reason why we should have included them in this order. This motivates the following construction.

**Definition 8** (($\sigma, \tau$)-FB persistence). *Let $\sigma, \tau$ be two permutations of the list $[n] = \{0, ..., n\}$. The $i$-th $(\sigma, \tau)-$**FB persistence** is the persistence diagram associated with the persistence module obtained by applying $H_i(-)$ to the sequence*

$$\emptyset = Y_{-1} \subset ... \subset Y_n = G = Z_0 \to Z_1 \to ... \to Z_n = *,$$

*where for $0 \le i \le n$, $Y_i = \bigcup_{j \le i} \text{IC}_{\sigma(j)}(G)$, and for $1 \le j \le n$, $Z_j = Z_{j-1}/(*_{j-1} \cup \text{IC}_{\tau(j)}(G))$ where $*_{j-1}$ is previously defined for $j - 1 \ge 1$, and $*_0$ is any point in $\text{IC}_{\tau(1)}(G)$.*

In other words, we apply Definition 3 to the sequence where we filtrate $G$ by spawning the intermediate complexes in the order $\sigma$, and then contracting $G$ in the order $\tau$. In the case where $\text{id}$ is the identity and $\text{re}$ is the reverse list bijection. FB-persistence is $(\text{id}, \text{re})$-FB-persistence.

A core motivation for Definition 8 comes from color-based filtrations (Example 2). Indeed, recall that degree filtration is a special case of Example 2. The perspective of coloring filtration is that we can spawn the vertices in any permutation of the set of possible degree values here, which may lead to more information and reduce bias for the model. Furthermore, to reduce bias, one could also argue why we should wait until the entire graph has been filtrated to start the contraction process.

Indeed, the whole process would still terminate to a point as long as we ensure the intermediate complex being contracted has appeared earlier in time, and this yields further flexibility in the process of switching back and forth between the forward steps and the backward steps (analogous to the imagery of an **hourglass**). This motivates us to the definition of **hourglass persistence**, which may be regarded as a "color-based version" of FB-persistence that also has a well-defined metric for persistence diagrams on different graphs.

**Definition 9.** *Let $f : X \to \mathbb{R}$ be a filtration function with associated intermediate complexes $\text{IC}(X_i)$. An **hourglass persistence diagram** is the persistence diagram of any sequence of inclusions and contractions, provided that $\text{IC}(X_i)$ is included in the sequence before it is contracted.*

Figure 4 gives an illustration of an example of a sequence of maps that occurs in hourglass persistence. To demonstrate its expressivity, we have:

**Proposition 2.** *Hourglass persistence is more expressive than FB-persistence (Figure 2(c)).*

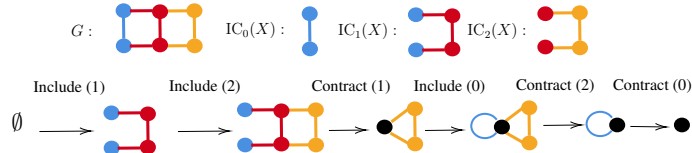

Figure 4: Example of a sequence of maps arising in the hourglass persistence for a filtration of $G$.

Hourglass persistence has promising potential to tackle a problem that PH-based methods sometimes struggle to scale on large inputs. This is because interleaving the filtration and contraction steps avoids the need to filtrate the entire space before starting contractions, which allows the total size of the space that appears in its entire lifetime to be bounded. Hourglass persistence hence gives an option for an **intermediate size control**. This gives an interesting tradeoff between runtime and expressivity, depending on when and how to contract. See Appendix E.4 for details.

## 4  RELATIONSHIP TO EXTENDED PERSISTENCE AND EXTENSIONS

Extended and FB-persistence are different as follows: For a filtration function $f$, extended persistence is obtained by considering a concatenation of PDs with filtrating by $f$ first and by $-f$ back. By Proposition 2.22 of Hatcher (2002), this is essentially akin to shrinking the superlevel sets $G^a := \{x \in V \cup E \mid f(x) \geq a\}$ as $a$ goes down, but the superlevel sets and the intermediate complexes coming from $f$ are in general quite different. We make this precise in Proposition 5.

We will now introduce a unifying perspective of both methods:

**Definition 10** (($f,g$)-FB Persistence)**.** *Let $f, g$ be two filtration functions on $G$ with $G_\bullet^f, G_\bullet^g$ being their induced filtrations, respectively. The $i$-th $(f, g)$-**FB Persistence** of $G$ is the persistence diagram associated with the sequence:*

$$\emptyset = G_{-1}^f \subset G_0^f \subset ... \subset G_n^f = G = G_0^g \to G_1^g \to ... \to G_m^g = *.$$

*where $\{G_i^f\}_{i \in \{-1,...,n\}}$ is the filtration of $G$ induced by $f$, and the sequence $G_\bullet^g := G_0^g \to G_1^g \to ... \to G_m^g = *$ is a sequence of contractions following the subgraphs that appear in the filtration induced by $g$. More precisely, $G_1^g := G/\operatorname{IC}_0(G, g)$, $G_2^g := G/(*_1 \cup \operatorname{IC}_1(G, g))$, $G_3^g := G/(*_2 \cup \operatorname{IC}_2(G, g))$, etc., where $*_i$ is the point representing the total contracted subcomplex from before.*

The idea of combining different filtration directions has also appeared in zigzag (Carlsson & de Silva, 2010) and bipath (Aoki et al., 2025) filtration. We explain in Appendix E how Definition 10 differs from the two. Definition 10 also generalizes extended persistence due to the following result. Note that the following is not difficult to prove; we only include it here for completeness.

**Proposition 3.** *The extended persistence of a filtration function $f$ (as defined in Section 2 of Yan et al. (2022)) has the same expressive power as $(f, -f)$-FB persistence.*

Note that the expressivity analysis of extended persistence had been carried out in Yan et al. (2025).

**Proposition 4.** *Let $f$ be a filtration function. There exist filtration functions $f_1, f_2$ such that the sequence of topological maps in $(f_1, f_2)$-FB persistence is exactly the sequence of topological maps in $(\sigma, \tau)$-FB persistence. If $f$ is vertex-based, we provide an explicit $O(N \log N)$ algorithm to compute $(f_1, f_2)$ that becomes linear time for FB-persistence (see Appendix B.3).*

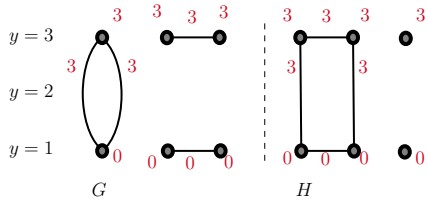

Figure 5: Example of graphs $G$ and $H$ with height filtration such that FB-persistence can differ but not $(f, -f)$-FB persistence.

Now we establish a clear separation between FB persistence and extended persistence.

**Proposition 5.** *There exist embedded graphs $G, H$ with height filtrations (Figure 5) where FB-persistence can tell them apart, but extended persistence cannot.*

Besides the choice of $-f$ or $f^b$, another candidate can be $f$ itself. Here we show that $(f, f)$-hourglass persistence does not bestow further information.

**Proposition 6.** *$(f, f)$-FB persistence has the same expressive power as forward-PH for $f$. For graphs, $(f, f)$-FB persistence can be computed by:*

- *Suppose there are $n$-steps in the filtration, a cycle born in filtration at $t = i$ dies at $t = n + i$.*
- *A cycle born in the contraction at $t = n + i$ corresponds exactly to the occurrence of a connected component at $t = i$ that will be merged into a pre-existing connected component in the filtration steps. The death time is $n + j$, and $j > i$ is the 1st time in filtration when the merge above occurs.*
- *The component death times are recorded as usual by keeping track of vertex representatives.*

## 5 EXTENSION TO SIMPLICIAL AND CELLULAR COMPLEXES

We now extend the framework to higher dimensions, viewing graphs as 1-dimensional simplicial complexes (for simple graphs) and 1-dimensional cell complexes. This extension can be readily integrated into simplicial and cellular networks (Bodnar et al., 2021a;b; Kim et al., 2020).

### 5.1 EXTENSION TO SIMPLICIAL COMPLEXES

Definition 3 works for any sequence of topological spaces, so our constructions may be defined directly. Unfortunately, the quotient of a simplicial complex may not be a simplicial complex without further triangulations.

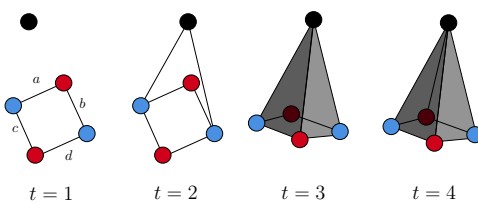

$t = 1$     $t = 2$     $t = 3$     $t = 4$

Figure 6: The contraction steps in Figure 2 interpreted as adding simplices instead.

We can however adapt a trick to conduct "simplicial quotients" (see Cohen-Steiner et al. (2006); Dey & Wang (2022)) as follows. Add a disjoint vertex $v_+$ to the simplicial complex at the beginning, and whenever a simplex $\sigma$ is asked to be contracted, one instead adds a simplex $[v_+, \sigma]$ to keep this as a filtration. On graphs, this means that one adds an edge from $v$ to $v_+$ for every vertex $v$ being contracted and a triangle $(v_1, v_2, v_+)$ for every edge $(v_1, v_2)$ being contracted (see Figure 6). One then computes the persistent diagram of this filtration instead.

**Proposition 7.** *For all methods defined in Section 3, 4, the simplicial quotient method will recover the $k$-th dimensional persistence diagrams for $k > 0$ but may differ on the level of $k = 0$. For $k = 0$, the pairs can be directly computed by keeping track of one vertex representative per component.*

### 5.2 EXTENSION TO CELLULAR COMPLEXES

The simplicial version adds more simplices to the set-up, which may make the computations more costly. We should try to exploit the contraction steps geometrically, as they would reduce the data. To do this, we would like to relax the collection of objects we are working with.

A **regular cell complex** (Hansen & Ghrist, 2019) is a generalization of simplicial complexes that adds more flexibility by allowing $X$ be built off of $k$-dimensional disks (called $k$-cells) as opposed to $k$-dimensional triangles. Intuitively, a cell complex is built out inductively by starting with the 0-cells (i.e., discrete points), attaching 1-cells to 0-cells, then attaching 2-cells to the complex, and so on. This process yields a poset structure on the cells where $\tau \leq \sigma$ indicates part of $\sigma$ is attached onto $\tau$. We refer to Section 2 of Bodnar et al. (2021a) for a thorough explanation.

Now we may extend a filtration function on a regular cell complex $X$ as follows.

**Definition 11.** *Let $\mathrm{Cell}(X)$ denote the collection of cells on $X$. A function $f : \mathrm{Cell}(X) \to \mathbb{R}$ is a filtration function if $f(\tau) \leq f(\sigma)$ for all $\tau \leq \sigma$ in $\mathrm{Cell}(X)$, which induces a filtration on $X$.*

An important feature of regular cell complexes is that they are closed under quotients of subcomplexes. Thus, every construction we discussed in Section 3 and 4 extends *mutatis mutandis* - that is, one can just replace the word "graph" with the word "cell complex".

## 6 STABILITY

Recall one motivation for "concatenating" inclusions and contractions was to compare metrics for diagrams on different spaces. We would still like to ensure the diagrams are stable if they are on the same space. This is a desired property as we would like perturbations in the input filtration to not affect the output drastically. To properly discuss (bottleneck) stability, however, we need to make a distinction between **combinatorial time** and **function time**.

So far, the persistence diagrams in our construction have all been combinatorial time - we use some function(s) to make a sequence of maps, and the pairs $(i, j)$ record the birth and death steps in the sequence. However, if we want a notion of bottleneck stability, we should work with function time - that is, we change the pair $(i, j)$ to the $(a_i, a_j)$.

When extending function time to include contractions, we observe that there are some subtleties in defining function time for $(f, g)$-FB persistence, because the functions $f$ and $g$ can be independent. It can be the case that the values of $g$ are mixed with the values of $f$ on the real line, but to define a function time we would like the values of $g$ to appear after the values of $f$. To get a well-defined notion of function time, we require $f$ and $g$ to both be positive, and that the corresponding function time of (a) an inclusion step at $i$ in $(f, g)$-FB persistence to be the $i$-th value of $f$ and (b) a contraction step at $n + i$ in $(f, g)$-FB persistence to $\max(f)+$ the $i$-th value of $g$.

> **Theorem 2** (Stability of $(f, g)$-FB persistence). *Let $X$ be a graph, simplicial or cellular complex. For 2 pairs of filtrations $(f, g), (f', g')$ on $X$, we have the following for all $i \geq 0$:*
>
> $$d_B(\mathrm{PH}_i^{FB}(X, f, g), \mathrm{PH}_i^{FB}(X, f', g')) \leq 2||f - f'||_\infty + ||g - g'||_\infty + |\max(f) - \max(f')|.$$

We remark that establishing an appropriate **function time** for hourglass persistence is challenging, because the function values for each step is not as canonical since we are interleaving the steps of contractions and inclusions. Hourglass persistence does satisfy the condition of what Chazal et al. (2009) called "tame", which has a **stability** in combinatorial time (see Theorem 4.4 therein).

## 7 ALGORITHM DESIGN AND EXPERIMENTS

We now present a practical algorithmic framework that supports any filtration scheme composed of a sequence of inclusions and contractions, assuming that all contraction intermediates have already appeared previously in the forward filtration.

### 7.1 FORWARD INCLUSION
WITH AUXILIARY BOOKKEEPING.

In addition to the standard union–find structure, which incrementally tracks connected-component memberships during the forward filtration, we maintain two further structures: (i) neighborhood information for the spanning forest being built, and (ii) a fundamental cycle basis over $\mathbb{F}_2$. Each new edge $e = (u, v)$ is then handled in two cases:

---

**Algorithm 1** FORWARDINCLUSION

1: **Input:** Filtration $f$; Graph $G$
2: **Output:** $\mathrm{PD}_0, \mathrm{PD}_1$, cycle basis $\mathcal{B}$, union–find UF
3: Initialize UF on $V$; $\mathrm{PD}_0, \mathrm{PD}_1, \mathcal{B} \leftarrow \emptyset$
4: Sort edges $e_1, \ldots, e_m$ by $f(e_j)$
5: **for** $j = 1..m$ **do**
6:     $(u, v) \leftarrow e_j$
7:     **if** UF. find$(u) = $ UF. find$(v)$ **then**
8:         // Cycle-creating edge
9:         Build $\gamma \in \{0, 1\}^m$ from $e_j$ and path $u \rightsquigarrow v$
10:         $\mathcal{B} \leftarrow \mathcal{B} \cup \{\gamma\}$; $\mathrm{PD}_1[\gamma] \leftarrow (f(e_j), \infty)$
11:     **else**
12:         $r_u \leftarrow$ UF. find$(u)$; $r_v \leftarrow$ UF. find$(v)$
13:         $y \leftarrow \arg\max_{r \in \{r_u, r_v\}} f(r)$
14:         $\mathrm{PD}_0 \leftarrow \mathrm{PD}_0 \cup \{(f(y), f(e_j))\}$
15:         UF. merge$(u, v)$
16:         // record mutual neighbors in spanning forest
17:         UF. nbrs $\cup \{u \leftrightarrow v\}$
18: **for** each root $r$ of UF **do** add $(f(r), \infty)$ to $\mathrm{PD}_0$
19: **return** $\mathrm{PD}_0, \mathrm{PD}_1, \mathcal{B}$, UF

---

**(1)** If $u$ and $v$ are in different components, $e$ is a *spanning-tree edge*. We update the union–find and record $u$ and $v$ as neighbors in the spanning forest, thereby incrementally extending the spanning structure maintained during the filtration.

**(2)** If $u$ and $v$ are already connected, $e$ is a *cycle-creating edge*. Together with the unique forest path $u \rightsquigarrow v$, this defines a new cycle $C_{k+1}$, given an existing basis $\{C_1, \ldots, C_k\}$. Since $e$ does not appear in any earlier cycle, $C_{k+1}$ is linearly independent of $\{C_1, \ldots, C_k\}$ in the cycle space over $\mathbb{F}_2$. The basis is thus extended to $\{C_1, \ldots, C_k, C_{k+1}\}$, and a new interval $(f(e), \infty)$ is added.

We denote by $PD_0$ and $PD_1$ the persistence diagrams of $H_0$ and $H_1$, respectively. Algorithm 1 describes the forward stage of our framework. Its input is a filtration function $f : V \cup E \to \mathbb{R}$ on a graph $G = (V, E)$, and its output consists of the persistence diagrams $PD_0$ and $PD_1$, a fundamental cycle basis $\mathcal{B}$ represented by indicator vectors in $\{0, 1\}^m$, and a union–find structure UF that maintains connected-component information and parent pointers for the spanning forest.

## 7.2 BACKWARD CONTRACTION WITH SUPERNODE BOOKKEEPING.

The backward stage uses a contraction function $g$ to order contractions. We maintain a *supernode* that accumulates contracted vertices; a vertex merges when scheduled by $g$, and an edge contracts once both endpoints reside in the supernode. Algorithm 2 summarizes the contraction process.

**Vertex contractions.** The contraction of a vertex into the supernode has two effects: **(1)** If the vertex belongs to a different connected component, the younger component is killed, and its $H_0$ interval is closed. **(2)** If the vertex lies in the same component, a new *supernode cycle* is created. Unlike forward cycles, these are not tied to any edge but arise from merging disconnected subgraphs of the same component.

**Edge contractions.** An edge is contracted once it becomes a self-loop on the supernode. Each such contraction kills one cycle: **(1)** If the edge participates in some forward cycle, we remove it from all cycle indicators in $\mathcal{B}$ and reduce the basis. If a younger cycle becomes dependent on older ones, it is killed, and its $H_1$ interval is closed. **(2)** If no forward cycle is removed, the contraction

---

**Algorithm 2** BACKWARDCONTRACTION

1: **Input:** contraction function $g$; persistence data $(PD_0, PD_1, \mathcal{B})$ and union–find UF from Alg. 1
2: **Output:** updated $PD_0, PD_1$ with finite deaths
3: Initialize supernode $S \leftarrow \emptyset$, stack $B$, list $L$
4: **for** elements $x \in V \cup E$ in order of $g(x)$ **do**
5:     **if** $x \in V$ **then**                    ▷ Node contraction
6:         $S \leftarrow S \cup \{x\}$
7:         **if** $UF.\text{find}(x) \neq UF.\text{find}(S)$ **then**
8:             // kill younger component $y$
9:             $PD_0 \leftarrow PD_0 \cup \{(f(y), g(x))\}$
10:        **else**
11:            $B.\text{push}(g(x))$    // Birth supernode cycle
12:        $UF.\text{merge}(x, S)$
13:    **else**                                   ▷ Edge contraction
14:        // $x = e = (u, v) \in E$ with $u, v \in S$
15:        remove $e$ from all $\gamma \in \mathcal{B}$; reduce basis
16:        **if** some $\gamma$ becomes dependent **then**
17:            // Close forward cycle
18:            $PD_1[\gamma] \leftarrow (\text{birth}(\gamma), g(x))$
19:            $\mathcal{B} \leftarrow \mathcal{B} \setminus \{\gamma\}$
20:        **else**
21:            // Close supernode cycle
22:            $\tau \leftarrow B.\text{pop}();\quad L \leftarrow L \cup \{(\tau, g(x))\}$
23: $PD_1 \leftarrow PD_1 \cup L$
24: **return** $PD_0, PD_1$

---

kills the most recent supernode cycle. In either case, the contraction assigns a finite death time to an $H_1$ interval, completing the bookkeeping of backward updates.

## 7.3 EXPERIMENTAL SETUP AND RESULTS

**Datasets.** We evaluate on four standard graph classification datasets (Morris et al., 2020): NCI109, PROTEINS, IMDB-BINARY, and NCI1 (Accuracy); a graph-regression dataset ZINC (Dwivedi et al., 2023) (MAE) and the OGBG-MOLHIV (Hu et al., 2020) molecular property dataset (AUC). All methods use the same GIN (Xu et al., 2019) or GCN (Kipf, 2016) backbone and the same DeepSets (Zaheer et al., 2017) pooling for encoding persistence diagrams.

**Baselines and ablations.** We compare five variants of persistence-based representations. **(i) PH** (Horn et al., 2021) uses a fixed vertex filtration and assigns each edge the maximum filtration of its endpoints, producing standard inclusion-based persistence. **(ii) RePHINE** (Immonen et al., 2023) learns both vertex and edge filtrations and, importantly, augments every 0D persistence point with the filtration value of the earliest incident edge, providing additional early-connectivity information. **(iii) Fwd-only** learns a general vertex–edge filtration but does *not* include RePHINE's augmentation, and disables all contractions; this isolates the effect of learning the forward filtration alone. **(iv) Bwd-only** learns only the contraction schedule, isolating the backward component of our design. **(v) Ours** implements the full learnable inclusion–contraction (forward–backward) framework, jointly learning both vertex/edge filtrations and contraction order.

**Observations.** Table 1 reports performance across all six benchmarks. We observe: **(1)** Our method achieves the best or second-best performance in 9 of 12 settings, underscoring the value of jointly learning both inclusion and contraction schedules. **(2) RePHINE** remains a strong baseline and consistently outperforms standard PH in nearly all cases, reflecting the benefit of learning

Table 1: Comparison of PH variants across six datasets using GIN and GCN backbones. Classification accuracy and AUC scores are reported in percentage (%, ↑) and ZINC regression evaluation in MAE (↓). Best and second-best results per row are shown in **bold** and underline, respectively.

| | Dataset | PH | RePHINE | Fwd-only | Bwd-only | Ours |
|---|---|---|---|---|---|---|
| **GIN** | NCI109 (Acc.%, ↑) | $76.76_{\pm0.40}$ | $\underline{77.89_{\pm1.19}}$ | $77.00_{\pm1.03}$ | $76.35_{\pm0.50}$ | $\mathbf{77.89_{\pm1.87}}$ |
| | PROTEINS (Acc.%, ↑) | $69.35_{\pm1.83}$ | $69.94_{\pm2.76}$ | $70.24_{\pm2.95}$ | $\underline{70.54_{\pm2.19}}$ | $\mathbf{73.51_{\pm1.11}}$ |
| | IMDB-B (Acc.%, ↑) | $68.67_{\pm1.25}$ | $70.67_{\pm0.94}$ | $\mathbf{74.67_{\pm0.47}}$ | $\underline{74.33_{\pm0.94}}$ | $72.00_{\pm2.16}$ |
| | NCI1 (Acc.%, ↑) | $79.24_{\pm1.74}$ | $78.75_{\pm2.55}$ | $76.72_{\pm1.13}$ | $75.75_{\pm0.94}$ | $\mathbf{81.27_{\pm0.00}}$ |
| | ZINC (MAE, ↓) | $0.43_{\pm0.01}$ | $\underline{0.41_{\pm0.01}}$ | $0.62_{\pm0.01}$ | $0.61_{\pm0.00}$ | $\mathbf{0.40_{\pm0.01}}$ |
| | MOLHIV (AUC%, ↑) | $\mathbf{74.34_{\pm4.57}}$ | $\underline{72.88_{\pm2.15}}$ | $70.00_{\pm3.11}$ | $70.59_{\pm1.83}$ | $72.34_{\pm0.74}$ |
| **GCN** | NCI109 (Acc.%, ↑) | $76.59_{\pm1.32}$ | $\mathbf{79.50_{\pm0.11}}$ | $71.91_{\pm0.52}$ | $74.58_{\pm0.71}$ | $\underline{75.87_{\pm0.89}}$ |
| | PROTEINS (Acc.%, ↑) | $70.54_{\pm0.73}$ | $68.75_{\pm2.53}$ | $69.35_{\pm1.83}$ | $\underline{70.54_{\pm1.46}}$ | $\mathbf{72.32_{\pm1.46}}$ |
| | IMDB-B (Acc.%, ↑) | $65.00_{\pm1.63}$ | $\mathbf{70.00_{\pm0.82}}$ | $62.67_{\pm3.30}$ | $64.33_{\pm3.30}$ | $\underline{68.00_{\pm2.16}}$ |
| | NCI1 (Acc.%, ↑) | $78.43_{\pm0.98}$ | $\mathbf{78.91_{\pm0.80}}$ | $75.59_{\pm1.00}$ | $76.24_{\pm1.74}$ | $\underline{78.67_{\pm1.69}}$ |
| | ZINC (MAE, ↓) | $0.49_{\pm0.02}$ | $\underline{0.46_{\pm0.01}}$ | $0.86_{\pm0.01}$ | $0.87_{\pm0.01}$ | $\mathbf{0.44_{\pm0.01}}$ |
| | MOLHIV (AUC%, ↑) | $75.12_{\pm0.68}$ | $\underline{75.40_{\pm0.53}}$ | $71.02_{\pm2.18}$ | $71.55_{\pm1.21}$ | $\mathbf{76.37_{\pm1.45}}$ |

both vertex and edge filtrations. **(3)** The ablations further clarify the role of each component: **Fwd-only**, which learns vertex–edge filtrations but disables contractions, typically improves over standard PH; **Bwd-only**, which learns only contraction order, also improves over PH on several datasets and performs roughly on par with **Fwd-only** overall, with no clear winner between them. **(4) RePHINE** significantly outperforms both **Fwd-only** and **Bwd-only** on ZINC and MOLHIV, highlighting the effectiveness of its additional 0D augmentation in capturing early connectivity structure. **(5)** Our full model almost always surpasses both ablations, demonstrating that inclusion and contraction encode complementary structural information, and that jointly learning them is essential for achieving the strongest performance across classification, regression, and molecular prediction tasks.

## 7.4 COMPARISON WITH EXTENDED PERSISTENCE

We further compare our framework to methods based on extended persistence, which incorporate both sublevel and superlevel information. PersLay (Carriere et al., 2020) implements this by first computing extended-persistence diagrams and then applying a learnable layer on top. In contrast, motivated by Proposition 3, which shows that extended persistence has the same expressive power as $(f, -f)$–FB persistence, we integrate extended persis-

Table 2: Accuracy (%) with GIN and GCN backbones. Best score per row is in **bold**.

| Dataset | GIN | | | GCN | | |
|---|---|---|---|---|---|---|
| | ExtP | PersLay | Ours | ExtP | PersLay | Ours |
| NCI109 | **78.21** | 68.28 | **78.21** | **77.48** | 68.28 | **77.48** |
| PROTEINS | **74.11** | 66.07 | 73.21 | **72.32** | 66.07 | **72.32** |
| IMDB-B | 63.00 | 70.00 | **73.00** | 66.00 | **70.00** | **70.00** |
| NCI1 | 78.59 | 68.86 | **81.51** | 80.05 | 68.86 | **80.78** |

tence directly into our forward–backward framework with $-f$ as a backward schedule. In Table 2, we observe: **(1)** Our $(f, -f)$ adaptation of extended persistence consistently outperforms PersLay under both GIN and GCN backbones, showing that embedding extended-persistence structure directly into the forward–backward framework is more effective than processing extended-persistence diagrams through a post-hoc learnable layer. **(2)** Our full $(f, g)$-forward–backward model further exceeds both PersLay and the extended variant, indicating that jointly learning inclusion and contraction provides expressive topological features beyond what extended persistence alone can capture.

## 8 CONCLUSION

We formulate *backward*, *(f,g)-forward–backward*, and *hourglass* persistence and analyze their expressive powers, certified by minimal witness graphs and constructive proofs, accompanied by algorithms that realize the general $(f, g)$ framework. These constructions extend to simplicial and cellular complexes and admit a functional stability guarantee, offering principled tools for inclusion–contraction schedules beyond the classical PH. While hourglass persistence satisfies combinatorial stability, establishing a functional stability result is a compelling direction for future work. Another interesting direction would be investigate the tradeoff between runtime and information of hourglass persistence.

## ETHICS STATEMENT

This paper presents work whose goal is to advance the interaction of topological descriptors with neural networks. Topological neural networks have many usage in the applied science, including medicine, finance, recommendation algorithms. There are many potential societal consequences of our work, none of which we feel must be specifically highlighted here.

## REPRODUCIBILITY STATEMENT

We provide the code at https://github.com/Aalto-QuML/Hourglass containing everything needed to reproduce our results: complete source code for all methods in PyTorch and C++; scripts to download datasets and generate deterministic splits with fixed seeds; exact configuration files and a pinned environment.yml with a README giving step-by-step commands to run all experiments; config files specifying hyperparameters and training protocols (optimizer, learning rate, batch size, epochs, scheduler, early stopping, and seeds) and a Jupyter notebook to generate results table. The appendix contains the full statements and proofs of all theorems, and the paper includes extensive diagrams that illustrate the separation examples and clarify our theoretical claims.

## ACKNOWLEDGMENTS

This research was conducted while MJ and IR were participating during the 2025 Aalto Science Institute international summer research programme at Aalto University. We are grateful to the anonymous program chair, area chairs and the reviewers for their constructive feedback and service. VG acknowledges the Academy of Finland (grant 342077), Saab-WASP (grant 411025), and the Jane and Aatos Erkko Foundation (grant 7001703) for their support. We thank Amauri H. Souza for helpful conversations on the code repository for Immonen et al. (2023). MJ would like to thank Mark Behrens and Erin Chambers for extensive helpful conversations and support. MJ would also like to thank Yuqin Kewang for helpful conversations on the content of this paper.

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

# Contraction and Hourglass Persistence for Learning on Graphs, Simplices, and Cells (Appendix)

## A    PROOFS FOR SECTION 3

In this section, we will provide the proofs for the results in Section 3.

*Proof of Proposition 1.* Let $G$ be a path graph on 4 vertices colored in the order R, B, B, R. Let $H$ be a star graph of 4 vertices such that the unique vertex with degree 3 has color B, and the other 3 vertices have color B, R, R (see Figure 3(a)). Consider a filtration of G and H by first spawning the induced subgraph on blue vertices, and then the induced subgraph on blue and red vertices. In other words, we have filtrations of the form

$$\emptyset \subset P_2 \subset G \text{ and } \emptyset \subset P_2 \subset H,$$

where $P_2$ is the path graph on two vertices, both colored blue. The inclusion-based PH of both filtrations are the same (see Figure 4 in Immonen et al. (2023)). However for backward-based PH, we observe that $G / \text{IC}_1(G)$ creates a cycle but $H / \text{IC}_1(H)$ does not create a cycle. Thus, they can be told apart in the contraction stage.

For the second part, consider the following pair of graphs in Figure 3(b), equipped with a vertex-based filtration using degree. In this case, we observe that the filtration of $G$ (left) and $H$ (right), respectively, is a sequence of discrete vertices until all the edges are spawned at the last time. This is because every edge is connected to a vertex of maximal degree equal to 3 for both $G$ and $H$. Thus, the backward persistence for both $G$ and $H$ would be to quotient out the entire graph, which cannot distinguish $G$ and $H$ since they have the same number of components and independent cycles. On the other hand, FB-persistence can tell $G$ and $H$ apart since in the filtration step spawning degree 1 vertices, 3 vertices are spawned for $G$ while 2 vertices are spawned for $H$. □

*Proof of Theorem 1.* We first see how the FB-persistence diagram can recover the forward persistence diagrams and the backward persistence diagrams. Indeed, at $n$ be the time the filtration completes and we are about to start contraction. The tuples that are born before or on time $n$ (which appears at the last step of the filtration) are of the form $(b, d)$ where $d$ is possibly greater than $n$. From here, we can recover the persistence diagram in the forward filtration as by sending each pair

$$(b, d), \text{ with } b \le n \mapsto \begin{cases} (b, d) \text{ if } d \le n \\ (b, \infty) \text{ if } d > n \end{cases}$$

where we note that if the pair $(b, d)$ has death after time $n$, then it must have died in the contraction step, which means that the feature lived to $\infty$ in the filtration step.

Similarly, we may obtain the backward persistence diagram by focusing on the persistent pairs $(b, d)$ such that $d > n$. We can recover them by the function

$$(b, d) \text{ with } d > n \mapsto \begin{cases} (0, d), \text{ if } b \le n \\ (b, d), \text{ if } b > n \end{cases} .$$

The reason why is because all the features that have not died yet at the start of contraction in FB-PH are the same as the features that are born at the initial step of the contraction scheme in backward PH. Thus, the elder rule applies to the same features when we decide what pairs to kill off. The only difference is that in backward PH, the features from the filtration steps all appeared at the same time, so it does not matter which one to kill if we do have to kill them, but for FB persistence we would need to be more careful. Evidently, though, this gives a straightforward reduction of backward PH from FB PH.

To see that FB PH is strictly more expressive than forward and backward PH, we consider the graphs $G$ and $H$ in Figure 3(c) using degree as a vertex-based filtration. Using the gudhi library (Project, 2025)'s persistence pairs computation, we can directly apply it to the following code in `Python`:

```
1  import gudhi
2
3  VG = [0, 1, 2, 3, 4, 5, 6, 7]
4  EG = [(0, 1), (1, 2), (1, 3), (2, 3), (3, 4), (4, 5), (4, 6), (4, 7), (6,
          7)]
5  G_values = [1, 3, 2, 3, 4, 1, 2, 2]
6
7  VH = [0, 1, 2, 3, 4, 5, 6, 7]
8  EH = [(0, 1), (1, 2), (1, 3), (2, 3), (3, 4), (3, 5), (5, 6), (5, 7), (6,
          7)]
9  H_values = [1, 3, 2, 4, 1, 3, 2, 2]
10
11 # Compute PDs for G
12 stG = gudhi.SimplexTree()
13 for i in range(0, len(VG)):
14     current_v = VG[i]
15     v_val = G_values[i]
16     stG.insert([current_v], filtration=v_val)
17
18 # Adding edges
19 for i in range(0, len(EG)):
20     current_e = EG[i]
21     e_val = max(G_values[current_e[0]],G_values[current_e[1]])
22     stG.insert(current_e, filtration=e_val)
23
24 stG.make_filtration_non_decreasing()
25 G_dgms = stG.persistence(min_persistence=-1, persistence_dim_max=True)
26 print(G_dgms)
27
28 # Compute PDs for H
29 stH = gudhi.SimplexTree()
30 for i in range(0, len(VH)):
31     current_v = VH[i]
32     v_val = H_values[i]
33     stH.insert([current_v], filtration=v_val)
34
35 # Adding edges
36 for i in range(0, len(EH)):
37     current_e = EH[i]
38     e_val = max(H_values[current_e[0]],H_values[current_e[1]])
39     stH.insert(current_e, filtration=e_val)
40
41 stH.make_filtration_non_decreasing()
42 H_dgms = stH.persistence(min_persistence=-1, persistence_dim_max=True)
43 print(H_dgms)
44
45 print(G_dgms == H_dgms)
```

The output would say that for graphs have the persistence diagrams of the form [(1, (3.0, inf)), (1, (4.0, inf)), (0, (1.0, inf)), (0, (1.0, 4.0)), (0, (2.0, 4.0)), (0, (2.0, 3.0)), (0, (2.0, 2.0)), (0, (3.0, 3.0)), (0, (3.0, 3.0)), (0, (4.0, 4.0))].

Now we show that they have the same backward persistence diagrams. At the initial time, they have the same number of connected components and cycles, so there are no distinctions between $G$ and $H$. In the first step, we contract the $\mathrm{IC}_3(G)$ or $\mathrm{IC}_3(H)$ (which is the closed star containing the unique vertex labeled with the value $4$). This is a connected subtree for both $G$ and $H$, so the contraction does not produce non-trivial persistence pairs. Then we contract $\mathrm{IC}_2(G)$ (or $\mathrm{IC}_2(H)$), which would kill a cycle on both sides. Since the birth time of the cycles are the same, we just mark one of the $(0, -)$ to die at the time. But by the time we contract $\mathrm{IC}_1(G)$ (or $\mathrm{IC}_1(H)$), we will kill the remaining cycle on both sides, which marks another one, and we are done after the last step. Thus, we see that they will have the same backward persistence diagram.

Now to see that $G$ and $H$ have different FB-persistence, we note that the birth time of the two cycles when contracting $\mathrm{IC}_2(G)$ (or $\mathrm{IC}_1(H)$) are different. For $G$, the cycle being contracted is born when the degree 3 vertices are spawned. For $H$, the cycle being contracted is born when the degree 4 vertices are spawned. This would create a different persistent pair and hence FB-persistence can tell them apart. $\square$

*Proof of Proposition 2.* Clearly FB-persistence is an example of hourglass persistence, so hourglass has at least as much expressivity as FB. Let $G, H$ be graphs constructed in the NetworkX library, with filtration function $f$ being degree-based vertex-level filtrations, as:

```
G = nx.from_edgelist([(0, 1), (1, 2), (2, 3), (1, 4), (4, 5)])
H = nx.from_edgelist([(0, 1), (1, 2), (1, 3), (3, 4), (4, 5)])
```

See Figure 3(d) for a visualization of the two graphs above. We wish to show they have the same FB-persistence but different hourglass persistence.

In this case, we note that the two graphs would have the same FB-persistence diagram with respect to $f$. Indeed, at $t = 1$, all the vertices labeled 1 are spawned for both $G$ and $H$, which gives three copies of the form $(1, -)$. At $t = 2$, the induced subgraph on vertices labeled 1 and 2 appears. We note here that on both $G$ and $H$, by the elder's rule, the two new vertices are both killed, thus both diagrams have the form
$$(1, -), (1, -), (1, -), (2, 2), \text{ and } (2, 2).$$
Finally, at $t = 3$, the vertex labeled 3 appears which makes both $G$ and $H$ connected at the step. The vertex labeled 3 dies at the same time, and two of the tuples labeled $(1, -)$ also dies. This gives
$$(1, -), (1, 1), (1, 1), (2, 2), (2, 2), \text{ and } (3, 3).$$

Now we enter the contraction steps. Indeed, we first contract $\mathrm{IC}_2(G)$ (or the corresponding version for $H$), evidently we are contracting isomorphic connected subtrees on $G$ and $H$, so the process would neither create non-trivial cycles or non-trivial steps. Then we contract $\mathrm{IC}_1(G)$ (or the corresponding version for $H$), for $H$ this is a subtree, so the contraction does not create new non-trivial tuples. For $G$, one might be tempted to think that this is contracting a disconnected graph. However, since we contracted $\mathrm{IC}_2(G)$ already, the two edges are actually connected here and forms a tree, so this also does not incur a change.

Finally, at the end, we remark the unique remaining tuple to die at $\infty$.
$$(1, \infty), (1, 1), (1, 1), (2, 2), (2, 2), \text{ and } (3, 3).$$

**Remark:** Note also that in the proof here we computed the persistence tuples in function time as opposed to combinatorial time (see Section 6 for a discussion), but this does not matter in terms of expressivity.

Finally, to see why hourglass persistence can tell them apart, we see that if we spawn $\mathrm{IC}_1(G)$ (resp. $\mathrm{IC}_1(H)$) first in the sequence, then they would incur two connected components for $G$ but only one for $H$, so hourglass persistence can tell them apart. $\square$

## B    PROOFS FOR SECTION 4

We split the discussion for the proofs for Section 4 into three parts:

1. In Appendix B.1, we establish a lemma that will be helpful in streamlining the proof of Proposition 6 in Appendix B.2. For completeness, we will also use a special case of this lemma to show how persistence modules can derive the usual interpretation of persistence pairs for a graph filtration. We will also introduce some concepts helpful in understanding the proof of Proposition 6.

2. In Appendix B.2, we prove Proposition 3, Proposition 5, and Proposition 6 (i.e., the expressivity parts of the section).

3. In Appendix B.3, we will explain the two algorithms for Proposition 4 and prove it.

### B.1    KEY LEMMA AND THE USUAL PERSISTENT PAIRS INTERPRETATION FOR GRAPH FILTRATIONS

Before we give a proof of Proposition 6, we first note that the description of death times for connected components holds on an elevated generality that both the proofs of Proposition 6 and (later) Proposition 7 would use, so we might as well extract it out as a formal lemma.

For completeness, we will also give a self-contained proof of how Definition 3 recovers the usual way to compute persistent pairs for graph filtrations (for instance, see Section 4 of Horn et al. (2021) or Algorithm 1 of Immonen et al. (2023)).

The lemma is as follows.

**Lemma 1.** *Let $X_\bullet = (\emptyset = X_{-1} \to X_0 \to X_1 \to ... \to X_N)$ be a sequence of inclusions and contractions of graphs (i.e., the set-up of hourglass persistence). Note that by necessity the first step $X_{-1} \to X_0$ has to include in a graph. The persistence pairs[1] for $H_0(X_\bullet)$ can be computed algorithmically as follows - the method of which we called "keeping track of one vertex representative per component":*

1. *In the filtration step $X_{-1} \to X_0$, we pick 1 vertex per connected component in $X_0$ and mark a tuple $(0, -)$ corresponding to that. We fix one of these vertices to be called the **supernode (denoted $*$)**, in the sense that any time we compare a vertex $v$ with $*$ to decide which one to kill off, we always kill off $v$.*
2. *For $i \geq 0$, if the step $X_i \to X_{i+1}$ is a filtration, then one treats this in algorithmic time as a procedure to spawn every new vertex that appears first and mark them as tuples $(i, -)$, and then spawn edges between them. For each edge between spawned, if an edge is joined between 2 different components represented by vertices $v$ and $w$, we mark the vertex born later to die at time $i$ and pick $v$ as the representative of the new component. (Since we do not focus on trivial deaths, we can discard them after this step). If the two vertices are born at the same time, we randomly pick 1 to kill off unless one of them is the supernode $*$, in which case we always kill off the other vertex.*
3. *For $i \geq 0$, if the step $X_i \to X_{i+1}$ is a contraction, say contracting a subgraph $G$. We loop through each connected component $X_j$ of $G$, we find the vertex $v_j$ representing the component $X_j$ belongs in and kill $v_j$ unless (1) it got killed already in some index $j' < j$ when looping through the $X_j$'s or (2) it is the supernode $*$.*
4. *After $i = N$, for every tuple not dead yet, we mark it to die at time $\infty$.*

**Remark 2.** *Note that the statement of Lemma 1 does not require $X_N$ to terminate with only the point $*$ left, so the case of hourglass persistence in Definition 9 is a special case in this lemma. A more specific case of Lemma 1 is when the maps are all inclusions. This will recover the usual procedure to compute the 0-dimensional PH of a graph filtration (as we just ignore Step 3).*

*Proof of Lemma 1.* Indeed let us recall from Remark 1 how one way to compute persistence diagrams are done. Let us interpret how the remark tells us how to compute the quantities here explicitly:

---

[1]Recall here we are not accounting for trivial deaths

1. We pick a non-zero vector $v$ of $H_0(X_0)$, and consider the sequence of linear subspaces generated by the iterated image of $v$ in $H_0(X_\bullet)$ until it becomes 0.

2. Then we remove this sequence of linear subspaces off of $H_0(X_\bullet)$. If there is still a non-zero vector $w$ in $H_k(X_0)$, we pick $w$ and repeat the process (if the linear map sends $w$ outside. Otherwise, we choose a non-zero vector from what is left in $H_k(X_1)$ (if it exists) and look at its iterated images ahead, and so on.

Let us also recall the following standard fact in algebraic topology - suppose $f : A \to B$ is a map of (say) cellular complexes, then the induced map $f_* : H_0(A; \mathbb{Z}/2) \to H_0(B; \mathbb{Z}/2)$ can be described exactly as follows - recall $H_0(A; \mathbb{Z}/2)$ and $H_0(B; \mathbb{Z}/2)$ are respectively isomorphic to the direct sum of copies over $\mathbb{Z}/2$ indexed over their path-connected components. For each path component $A_i$, let $1_{A_i}$ be the unique non-zero vector representing that component. The continuous image of a path-connected component is path-connected, so $f(A_i)$ lies in a path-connected component of $B$, say $B_j$, then the map described above sends $f_*(1_{A_i}) = 1_{B_j}$.

**Remark 3.** *This description might deceptively suggest that $f_*$ is not the zero map on all of $H_0(A; \mathbb{Z}/2)$, but this is clearly not the case, it just happens that we chose a convenient basis on both sides such that $f_*$ is not zero on the chosen basis vectors. The subtlety in the persistence calculation is that we want to choose the basis in the $H_0(-)$ of the next space consistently with the basis of the previous one to nicely compute the persistence pairs.*

By the remark in the line above, though, we see that it follows that there exists a non-zero vector $v_1 \in H_0(X_0)$ (as long as $X_0$ is not empty, by requirement) such that the successive images of $v$ are never 0 when passing through $H_0(X_\bullet)$, thanks to the convenient choice of basis, and $v$ can in fact come from representing some connected component. We then choose $v$ here to represent the **supernode** $*$. In the next step of the remark, we then remove the subspace generated by successive images of $*$ off $H_0(X_0)$, and if the next vector we pick lands in this subspace, we consider it dead. Note that on the level of $H_0$, a vector $1_C \in H_0(X_i)$ representing a component could land in this subspace at $H_0(X_{i+1})$ if and only if $f_i(C)$ belongs in the component of the supernode at time $i + 1$. Thus, we see that asking a vector representing $1_C$ to die if it enters this subspace corresponds **exactly** to the property of the supernode.

Now we proceed by induction and suppose the $i$-th vector $v_i$ we pick following the remark (1) represents a component, (2) gives the correct persistence pair according to the outline of this lemma, and (3) has been picked according to the rules of Remark 1. Now we wish to show we can pick the $i + 1$-th vector $v_{i+1}$ such that they still satisfy (1) and (2). Now we would stop if we have ran out of vectors to pick, so there is at least some non-zero vector in the complement of the subspace generated by vectors $v_1, ..., v_i$ and their iterated images (call this $W_i$ for convenience). So there exists some vector $v_{i+1}$ that represents a component $C$ and say $v_{i+1} \in H_0(X_j)$.

By assumption, $v_{i+1}$ cannot be in the image of any previous vector, so the component $C$ must be a new component that appears from $X_{j-1} \to X_j$ (by the description of what the correspondent map in $H_0$ is like). Now we have two descriptions of how $C$ gets killed:

1. $C$ is killed in the sense of this lemma if and only if when (a) it merges with an earlier component during filtration or (b) a subgraph of $C$ was asked to be contracted in the future.
2. Remark 1 says that $C$ is killed if and only if the iterated image of $1_C$ gets landed in $W_i$.

We will go from the remark and see why it is equivalent to the first itme. Based on the description of how the induced map on $H_0(-)$ behaves, we see that $1_C$ can land into $W_i$ at time $j' > j$ if and only if there is some $0 \le k \le i$ such that $f_{j'-1} \circ ... \circ f_j(C)$ and $f_{j'-1} \circ ... \circ f_j(C_k)$ represents the same connected component, where $C_k$ is the component component $v_k$ represents at the time it was born.

Without loss, we can choose $j'$ to be the first time step $> j$ such that $(f_{j'-1})_* \circ ... \circ (f_j)_*(1_C)$ is in $W_i$ and $k$ to be the first $v_k$ whose component $C$ merges with. Then it follows that in the step $f_{j'-1} : X_{j'-1} \to X_{j'}$, $C$ dies by mergining into $f_{j'-1} \circ ... \circ f_j(C_k)$. If $f_{j'-1}$ is a filtration, this can

only happen if some edge is spawned between them. If $f_{j'-1}$ is a contraction, this can only happen if $k = 0$ (i.e., it goes into the supernode). Thus, we see that these are exactly the two conditions proposed by the scenario in the lemma. Furthermore, the birth and death times proposed by the lemma and Remark 1 agree.

We then induct repeatedly and conclude the proof. $\qquad\square$

As noted earlier, Lemma 1 recovers the usual algorithm to compute persistence pairs for graph filtrations on the 0th dimension. We now explain how to obtain the one for the 1st dimension from Definition 3. This will follow from the following more general fact.

**Lemma 2.** *Let $X$ be a (finite) cell complex of dimension $D$ and $X_\bullet := \emptyset = X_{-1} \to X_0 \to ... \to X_n = X$ be a filtration of $X$ by cellular sub-complexes, then the persistence pairs of $H_D(X_\bullet)$ may be computed as follows:*

1. *At $t = 0$, we instantiate $\dim H_D(X_0)$ many tuples of the form $(0, \infty)$.*

2. *For $t > 0$, we instantie $\dim H_D(X_t) - \dim H_D(X_{t-1})$ many tuples of the form $(t, \infty)$.*

3. *We end at $t = n$.*

We first look at how Lemma 2 specializes for graphs, before proving it. When $D = 1$, a 1-dimensional cell complex is the same as a graph, and the lemma recovers the usual way to calculate 1-dimensional PH's.

Indeed, $H_1(-)$ of a graph corresponds exactly to its list of independent cycles. For completeness, by "independent cycles", we mean the following interpretation:

**Definition 12.** *Let $C, C'$ be two cycles of the graph $G$. The $C \boxtimes C'$ as the XOR of $E(C)$ and $E(C')$ (i.e., their symmetric difference). For a list of cycles $D_1, ..., D_n$, the XOR span of the list is the following set:*
$$\{D_{i_1} \boxtimes D_{i_2} \boxtimes ... \boxtimes D_{i_k} \,|\, i_1, i_2, ..., i_k \subseteq \{1, ..., n\}\}$$

**Definition 13.** *Let $C_1, C_2, ..., C_k$ be a list of cycles of $G$. The list is a list of **independent cycles** if for any $C_i$, $C_i$ is not contained in the XOR span of $C_1, ..., C_{i-1}, C_{i+1}, ..., C_k$*

The following is a well-known interpretation of $H_1(-)$ of a graph and can be interpreted as the definition.

**Fact:** The maximal list of independent cycles on $G$ forms a basis for $H_1(G)$.

The usual algorithm for 1st dimensional persistence diagrams of a graph filtration keeps track of vertex representatives for the components, and they mark the birth-time of cycles of the form $(t, \infty)$ at filtration time $t$ for each edge drawn from a component to itself at time $t$.

**Proposition 8.** *Lemma 2 recovers the usual algorithm described above.*

*Proof.* Let us start from the algorithm side and work to Definition 3. Indeed, each $(t, \infty)$ corresponds to an edge $e$ that is born in $G_t$ (the subgraph at time $t$). Choose $C_e$ to be a cycle in $G_t$ containing the edge $e$, we choose this for every such edge arising above.

By Lemma 2, it suffices for us to check that for each time step $T$, $H_1(G_t)$ has a basis being $\{C_e\}_{e \in E}$, where $E$ is the collection of cycle-creating edges born in time $\leq T$. We first check that this is linearly independent. Indeed, we choose a total ordering $\leq$ on $E$ such that $e_1 < e_2$ if $e_1$ appeared earlier than $e_2$ in the algorithm, (even if they are born at the same filtration time, there is an ordering of which one is born first in algorithmic time). Under this total ordering, we rewrite the elements of $E$ into $e^1, e^2, ..., e^N$.

Clearly the list $\{C_{e^1}\}$ is independent, since $C_{e^1}$ is a non-trivial cycle. Suppose by induction $\{C_{e^1}, ..., C_{e^k}\}$ is independent, we wish to show adding $C_{e^{k+1}}$ into the list remains independent.

Indeed, the XOR span of any sublist of $\{C_{e^1}, ..., C_{e^k}\}$ is always contained in the union of edges of $C_{e^1}, ..., C_{e^k}$'s, which does not contain the edge $e^{k+1}$ because of the total ordering we picked. Thus, $C_{e^{k+1}}$, which contains $e^{k+1}$ by construction, is not in the XOR span. Thus, we have verified this is a **list of independent cycles**.

Now, to show that this is a maximal list. We observe that removing the set $E$ from $G_t$ will turn $G_t$ into a forest. This implies that there are no additional cycles left, which concludes the proof. □

Now we will prove Lemma 2.

*Proof of Lemma 2.* This lemma follows from the fact that the induced map $H_D(X_t) \to H_D(X_{t+1})$ by inclusion has to be injective for all $t$. Indeed, this comes from the long exact sequence in homology for the pair $(X_{t+1}, X_t)$ (see Theorem 2.16 of Hatcher (2002)), since $H_{D+1}(X_{t+1}, X_t)$ is evidently zero as $X_{t+1}$ and $X_t$ are both at most $D$-dimensional. Since this is injective, Remark 1 tells us that a chosen component according to the steps of the remark can never merge into a pre-existing component, so all the vectors picked survive to $\infty$. Once we move from $X_t$ to $X_{t+1}$, the new pairs that are created are exactly $\dim H_D(X_{t+1}) - \dim H_D(X_t)$ many pairs of the form $(t+1, \infty)$. This concludes the proof. □

## B.2 PROOFS FOR THE EXPRESSIVITY PARTS OF SECTION 4

*Proof of Proposition 3.* As defined in Section 2 of Yan et al. (2022), the extended persistence of a filtration function $f$ is equivalent to the persistent pairs associated to the persistence module:

$$0 = H(G_{-\infty}) \to ...H(G_a) \to H(G) = H(G, G^\infty) \to ...H(G, G^a) \to H(G, G^{-\infty}).$$

Here, $G_a$ means $\{x \in G \mid f(x) \leq a\}$ and $G^a$ means $\{x \in G \mid f(x) \geq a\}$, and $H$ denotes either $H_0$ or $H_1$. Here we note that the paper Yan et al. (2022) wrote $\emptyset = H(G_{-\infty}) = H(\emptyset)$, but some prefer the convention that $H(-)$ of the empty set is 0, as opposed to the empty set. One motivating reason is that the empty set is not a vector space. This does not affect the persistence pairs produced since they start at the non-zero parts.

Although $a$ is indexed over the entire extended real numbers $[-\infty, +\infty]$, since $f$ takes only finitely many values (or, if $f$ is a real valued function taken on a graph $G$, viewed as a non-discrete topological space, the topological changes only occur when a vertex is spawned), the persistence module reduces to a finite length persistence module of the form

$$0 \to H(G_{a_0}) \to ... \to H(G_{a_n}) = H(G) \to H(G, G^{a_n}) \to ... \to H(G, G^{a_0}),$$

where $a_0 < ... < a_n$ is the sequence of filtration values of $f$. By Proposition 2.22 of Hatcher (2002), there is a morphism of persistence modules of the form:

$$
\begin{array}{ccccccccc}
0 & \longrightarrow & H(G_{a_0}) & \longrightarrow & ... & \longrightarrow & H(G_{a_n}) & \longrightarrow & H(G, G^{a_n}) & \longrightarrow & ... & \longrightarrow & H(G, G^{a_0}) \\
& & \downarrow_{=} & & \downarrow_{=} & & \downarrow_{=} & & \downarrow & & & & \downarrow \\
0 & \longrightarrow & H(G_{a_0}) & \longrightarrow & ... & \longrightarrow & H(G_{a_n}) & \longrightarrow & H(G/G^{a_n}) & \longrightarrow & ... & \longrightarrow & H(G/G^{a_0})
\end{array}
$$

where the second row is analogous to the construction of $(f, g)$-FB persistence we did. Observe we can rewrite $G^{a_i}$ as $\{x \in G \mid -f(x) \leq -a_i\}$. By and the successive union of the intermediate complexes arising in the definition of $(f, -f)$-FB persistence up to step $i$ is exactly the same as $G^{a_i}$, so the persistence module of the second row arises exactly from the $(f, -f)$-FB persistence.

Proposition 2.22 of Hatcher (2002) also implies that the morphism above is an isomorphism of persistence modules if $H$ is $H_1(-)$, so they will have the same 1-dimensional persistent diagrams.

If $H$ is $H_0(-)$, then the maps $H(G, G^{a_i}) \to H(G/G^{a_i})$ injects onto a direct summand $\tilde{H}(G/G^{a_i})$ of $H(G/G^{a_i})$ such that $H(G/G^{a_i}) = \tilde{H}(G/G^{a_i}) \oplus \mathbb{Z}/2$. On the other hand, there is a very explicit interpretation on what the generator of the $\mathbb{Z}/2$ summand is - it is exactly the image of the chosen **supernode** (see Lemma 1). The difference here is that the supernode in the second row becomes the

only vertex feature that does not die, of the form $(0, \infty)$, and the supernode in the first row becomes the vertex feature of the form $(0, d)$ where $d$ achieves the maximum death time among all vertices of the graph (concretely, it is the first $i$ (from $n$ to 0) such that $G/G^{a_i}$ is a connected graph). By the decision rule we imposed on the supernode, we see that the first and second row agree on all 0-dimensional persistence pairs except for the supernode, but clearly we can invert from one to another.

This concludes the discussion that they have the same expressivity. □

*Proof of Proposition 5.* We show that the they have the same extended persistence but different FB persistence.

Indeed, consider the pair of graphs $G$ and $H$ from Figure 5 with the filtration function $f$ being the vertex-based filtration function induced by the height function on the vertices. For convenience, we also redraw the two graphs below as:

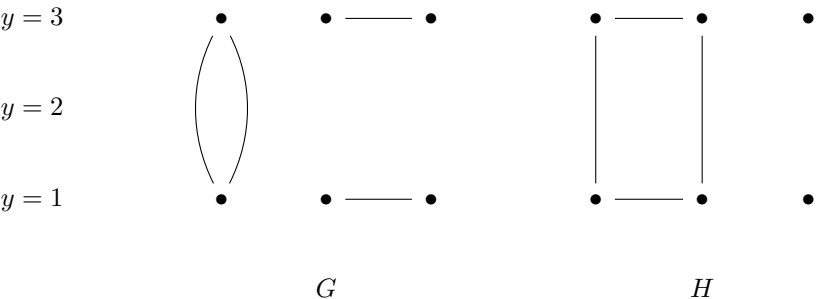

$$G \qquad\qquad\qquad\qquad H$$

In the forward time, we observe that the respective filtrations for $G$ and $H$ only change twice as $\emptyset = G_{-1} \subset G_0 \subset G_1 = G$ and $\emptyset = H_{-1} \subset H_0 \subset H_1 = H$, where $G_0$ and $H_0$ are identical. The same number of vertex births/deaths and cycle births occur at $y = 3$, and hence $G$ and $H$ cannot be told apart in forward time. If we apply the backward contraction with respect to $-f$ (i.e., extended persistence), then at $y = 3$, the subgraph being contracted is the same, and they both only kill 1 connected component. No changes happen until we get back to $y = 0$, but by then the entire remaining graphs are contracted, and no differences are detected. Thus, we conclude that forward with respect to $f$ + backward with respect to $-f$ cannot tell apart $G$ and $H$.

On the other hand, we observe that FB-persistence can clearly tell them apart. This is because $\mathrm{IC}_1(G)$ has a cycle and $\mathrm{IC}_1(H)$ does not, so contracting $G$ by $\mathrm{IC}_1(G)$ kills a cycle but contracting $H$ by $\mathrm{IC}_1(H)$ does not. □

*Proof of Proposition 6.* From a similar proof to that of Theorem 1, we know that $(f, f)$-FB has at least the same expressivity as forward PH with $f$. Note that the explicit description of how to compute $(f, f)$-FB persistence means that it is actually a function of the filtraiton function $f$, so if the explicit description holds then forward PH with $f$ is at least as expressive as $(f, f)$-FB persistence, which will show they have the same expressive power.

Thus, it suffices for us to verify this explicit description. The case for connected components is resolved by Lemma 1 already, so we will examine how to compute the case of cycles.

By a similar procedure to the explaination in Theorem 1 (essentially due to elder's rule), the cycles that appear in the contraction steps for $(f, f)$-FB persistence are exactly the cycles (that are not born at the beginning) in the backward persistence diagram that contracts the intermediate complexes in the order $\mathrm{IC}_0(G, f), \mathrm{IC}_1(G, f), ..., \mathrm{IC}_n(G, f)$. Note that a cycle can be born in the contraction stage if and only if we are being asked to contract two components that belong to the same connected component of the entire graph $G$ (note that, without loss in algorithmic time, we can always contract two components at a time).

Since the intermediate complexes being contracted are in the same order they appear in the filtration, we see that the birth of cycles in the contraction steps corresponds exactly to the appearance of

connected components in filtraiton that will be merged to connected components that are born earlier. The death time of such cycle also corresponds to when the two components actually merge. Perhaps one way to see this is to note that in the simplicial quotient interpretation (see Section 5), this fills in a list of triangles to the disjoint base point $v_+$ on top of a path connects the two vertex representatives. Prior to the filling, the two vertex representatives would each have an edge to $v_+$ (which is a cycle since they are in the same component for the entire graph $G$). The filling of the triangles here would kill the cycle. This proves the case for cycles born in contraction.

Finally, for the cycles born in filtration, we would like to predict their death time in contraction. Recall from the proof of Proposition 8 that there is an explicit description of the compatible basis of $H_1(G_\bullet^{\mathrm{fil}})$, where $G_\bullet^{\mathrm{fil}}$ denotes the sequence of inclusions of subgraphs induced by the filtration function $f$. The basis of $H_1(G)$ correspond to cycle-creating edges, and an explicit ordered basis of independent cycles $C_1 < ... < C_N$ can be found by choosing cycles $C_i$ that contain the cycle-creating edges $e_i$ in an inductive way (see the proof of Proposition 8 for more details). The upshot is that this would give a linearly independent list of cycles because the cycle-creating edge $C_i$ contains is not contained in $C_j$ for $j < i$.

In the contraction step, we observe that a necessary condition for the list $\{C_1, ..., C_N\}$ to degenerate (i.e., become linearly dependent) is when some cycle creating edge $e_i$ is contracted. Furthermore, if $e_i$ is the last edge in the cycle $C_i$ to be contracted, the cycle $C_i$ would die right there. The diffcult with general $(f, g)$-FB persistence is that, due to the arbitrariness of $g$, often times the cycle creating edge $e_i$ is not the last edge being contracted in $C_i$. In the case of $(f, f)$-FB persistence, however, we observe that we can always choose $e_i$ to die last in algorithmic time. Thus, this shows that each cycle $C_i$ born at time $k$ dies exactly at time $n + k$ (when the same cycle is being asked to be contracted with respect to $f$). □

### B.3 Algorithm to Compute $(\sigma, \tau)$-Forward-Backward Filtrations

In this section, we will prove Proposition 4 and explain the algorithms behind in the case when $f$ is vertex-based. We also remark that a similar algorithm exists in the arbitrary case when $f$ is not vertex-based, just one also has to carefully permute the edge based values. We chose to present the case when $f$ is vertex-based for simplicity.

Before we start explaining the algorithm, we first explain why $(f_1, f_2)$ exists in general. Indeed, this will follow immediately from the following more general fact.

**Lemma 3.** *Let $G$ be a graph and $G_\bullet = (\emptyset = G_{-1} \subset G_0 \subset G_2 \subset ... \subset G_n)$ be a strict inclusion of subgraphs on $G$, then there exists a filtration function $f : G \to \mathbb{R}$ with filtration values $a_0 < ... < a_n$ such that $f^{-1}((-\infty, a_i]) = G_{a_i}$.*

*Proof.* For each vertex or edge $x \in G$, we define $f(x)$ to be $\min_{i \in \{0,1,...,n\}} x \in G_i$. Since $G_\bullet$ is an inclusion of subgraphs, an edge $e$ cannot appear earlier than the vertices that support it, so it follows that $f$ is a filtration function. This concludes the proof. □

From now on in this subsection, we assume that $f$ is vertex-based for simplicity. Let us first look at the algorithm for FB-persistence specifically and see why it is only linear time.

**Proposition 9.** *Let $f$ be a filtration function, then there exists a filtration function $f^b$ such that the sequence of topological maps in $(f, f^b)$-persistence is the same as the ones in FB-persistence with respect to $f$. Furthermore, Algorithm 3 computes $f^b$ in with a runtime of $O(|V| + |G|)$.*

*Proof.* To show that Algorithm 3 correctly computes $f^b$, it suffices for us to show that $\mathrm{IC}_i(G; f^b)$ is exactly $\mathrm{IC}_{n-i}(G; f)$. Indeed, the intermediate complexes produced for the pair $(G, f)$ are "upward closed" in the sense that the edges of $\mathrm{IC}_i(G; f)$ all have the same value $a_i$, but the value of vertices are in general only $\leq a_i$.

If we want $\mathrm{IC}_n(G; f)$ to be the first intermediate complex that appears in the filtration $f^b$, we would want to relabel all of its vertices to the maximal value $a_n$, which is precisely what the algorithm

---

**Algorithm 3** Computing the Backward Filtration Function

**Input → Output:** The graph $G$ and filtration function $f$ → The function $f^b$

1: $f^b_{\text{vertex}} \leftarrow \{v : -\infty \mid \text{for each } v \in V(G)\}$      ▷ Initialize $f^b$ on vertices with values in $-\infty$.
2: **for** edge $e = (v, w)$ in $E(G)$ **do**
3:      $f^b_{\text{vertex}}[v] \leftarrow \max(f(e), f^b_{\text{vertex}}[v])$.
4: **for** vertex $v$ in $V(G)$ **do**
5:      **if** $f^b_{\text{vertex}}[v] = -\infty$ **then**
6:          $f^b_{\text{vertex}}[v] \leftarrow f(v)$      ▷ Mark isolated vertices to their value under $f$.
7: $(f^b_{\text{vertex}}, f^b_{\text{edge}}) \leftarrow (\text{map}(x \mapsto -x; f^b_{\text{vertex}}), \text{map}(x \mapsto -x; f_{\text{edge}}))$.
8: **return** $(f^b_{\text{vertex}}, f^b_{\text{edge}})$.

---

does. Since $f$ is vertex-based, we only need to do this on vertices and we can modify the edges later.

If we want $\text{IC}_{n-1}(G; f)$ to be the second intermediate complex that appears in the filtration $f^b$, we would want all vertices that have not been marked $a_n$ already to be marked $a_{n-1}$. This amounts to a maximality comparison in the for loop in the algorithm.

If we want $\text{IC}_{n-2}(G; f)$ to be the third that appears, we similarly want to label all vertices not marked $a_n, a_{n-1}$ yet to be marked as such, so repeating this process yields the correctness of the algorithm.

The algorithm has runtime $O(|V| + |G|)$ since it only requires looping through the vertex set and the edge set linearly.     □

Now we move on to Proposition 4. In fact, Algorithm 3 before is a special case of the algorithms for this. In order to create a backward-filtration function that contracts in a permutation of the intended order specified by a permutation $\sigma$, we observe that Algorithm 3 would actually hold if the max function is operated with respect to an "ordering given by $\sigma$" as opposed to the natural ordering of the reals.

Instead of changing the ordering system of the reals in our algorithm, we will instead change the filtration function $f$ inside the parameter of the max function to this ordering. To do this, we define the following variant of $f$:

**Definition 14.** *Let $f : G \to \mathbb{R}$ be a filtration function with filtration values $a_1 < a_2 < ... < a_n$. Let $\sigma$ be a permutation of the list $\{1, ..., n\}$, we define $\sigma \cdot f$ as the function*

$$\sigma \cdot f : G \to \mathbb{R}, \quad \sigma \cdot f(x) = a_{\sigma(i)} \text{ if } f(x) = a_i,$$

*where $x$ is either a vertex or an edge.*

---

**Algorithm 4** Computing the $\tau$-Backward Filtration Function

**Input → Output:** The graph $G$, function $f$, permutation $\tau^{-1}$ → The function $f^\tau$

1: $g \leftarrow (\text{re} \circ \tau^{-1}) \cdot f$
2: $f^\tau_{\text{vertex}} \leftarrow \{v : -\infty \mid \text{for each } v \in V(G)\}$      ▷ Initialize $f^\tau$ on vertices with values in $-\infty$.
3: **for** edge $e = (v, w)$ in $E(G)$ **do**
4:      $f^\tau_{\text{vertex}}[v] \leftarrow \max(g(e), f^\tau_{\text{vertex}}[v])$.
5: **for** vertex $v$ in $V(G)$ **do**
6:      **if** $f^\tau_{\text{vertex}}[v] = -\infty$ **then**
7:          $f^\tau_{\text{vertex}}[v] \leftarrow g(v)$      ▷ Mark isolated vertices to their value under $f$.
8: $f^\tau_{\text{vertex}} \leftarrow \text{map}(x \mapsto -x; f^\tau_{\text{vertex}})$.
9: $f^\tau_{\text{edge}} \leftarrow \text{map}(x \mapsto -x; g_{\text{edge}})$.      ▷ $g_{\text{edge}}$ is the function $g$ on $E(G)$.
10: **return** $(f^\tau_{\text{vertex}}, f^\tau_{\text{edge}})$.

---

In practice, a permutation $\sigma$ can be realized as a dictionary with entry being $i$ and output being $\sigma(i)$. We can use this to make an associated dictionary whose entry is $a_i$ and output is $a_{\sigma(i)}$. Depending on how the filtration function $f$ is represented as a data structure, this may require a sorting of the list of filtration values if the values are not sorted already. We can then produce $\sigma \cdot f$ by a linear scan through the entries of $f$ and swap out its output using the associated dictionary.

A slight modification of Algorithm 3 now produces Algorithm 4. In the special case when $\tau = \mathrm{re}$ (the reverse list permutation), this will recover Algorithm 3. This gives $f_2$ as requested in Proposition 4, but note that plugging $\sigma$ into $\tau$ also gives the desired $f_1$ in the proposition. A slight modification of the proof for Algorithm 3 will ensure the correctness of the algorithm here. Because we possibly may need to sort a list, this would incur a worst-case runtime of $O(N \log N)$, where $N = |V(G)| + |E(G)|$.

## C  PROOFS FOR SECTION 5

*Proof of Proposition 7.* Let $Y$ be a simplicial complex, and let $Y_\bullet$ be a sequence of inclusions and contractions in some arbitrary order, of the form:

$$\emptyset = Y_{-1} \to Y_0 \to Y_1 \to ... \to Y_m \to Y_{m+1} = *.$$

Using the simplicial quotient method, we adjoin a disjoint base point $v_+$ to $Y$ and also form a sequence of inclusions $Z_\bullet$ of the form

$$v_+ \to Z_0 \to Z_1 \to ... \to Z_m \to Z_{m+1}.$$

Taking $H_k(-)$ gives the sequence

$$H_k(Z_\bullet) : 0 \to H_k(Z_0) \to H_k(Z_1) \to ... \to H_k(Z_m) \to H_k(Z_{m+1}).$$

Fix $k > 0$, we also let $H_k(Y_\bullet)$ be the $k$-th persistent module for this from Definition 3.

Now for each $Z_i$, let $C_i$ be the closed star of the vertex $v_+$ (i.e., the union of all simplices containing $v_+$ and their faces). Observe that $Y_i$ is isomorphic to the quotient $Z_i/C_i$. We also note that $C_i$ is actually contractible - indeed, in the standard geometric realization of $Z_i$, there is an explicit straight line homotopy based on line segments from the simplicial link of $v_+$ to $v_+$ from construction, which shows that $C_i$'s are contractible.

It is a general fact that for a simplicial complex $K$ and contractible subcomplex $K'$, the quotient map $K \to K/K'$ is a homotopy equivalence (see Proposition 0.17 of Hatcher (2002)).

$$
\begin{array}{ccccccccc}
0 & \longrightarrow & H_k(Z_0) & \longrightarrow & H_k(Z_1) & \longrightarrow & ... & \longrightarrow & H_k(Z_m) & \longrightarrow & H_k(Z_{m+1}) = 0 \\
& & \cong\downarrow & & \cong\downarrow & & \cong & & \cong\downarrow & & \parallel\downarrow \\
0 & \longrightarrow & H_k(Z_0/C_0) & \longrightarrow & H_k(Z_1/C_1) & \longrightarrow & ... & \longrightarrow & H_k(Z_m/C_m) & \longrightarrow & H_k(\{*\}) = 0 \\
& & \cong\downarrow & & \cong\downarrow & & \cong & & \cong\downarrow & & \parallel\downarrow \\
0 & \longrightarrow & H_k(Y_0) & \longrightarrow & H_k(Y_1) & \longrightarrow & ... & \longrightarrow & H_k(Y_m) & \longrightarrow & H_k(\{*\}) = 0
\end{array}
$$

Here $H_k(Z_{m+1}) = 0$ because $C_{m+1} = Z_{m+1}$ in this case. Thus, we have an isomorphism of persistence modules between $H_k(Z_\bullet)$ and $H_k(Y_\bullet)$, so they have the same persistence diagrams.

To see what goes wrong in the case $k = 0$, we observe that the placement of the node $v_+$ can affect the birth / death time of vertices since $v_+$ is now the oldest node, so the birth times of the vertices would shift. This can be remedied, for FB-persistence or $(\sigma, \tau)$-persistence, where we instead place $v_+$ to spawn after filtration finishes and before contraction begins. However, this does not work for hourglass persistence, since we can contract before all the filtration finishes.

Thus, we would like to consider a different method, as described in the proposition. For a simplicial complex $K$, we write $(K)^1$ to denote the 1-skeleton of the simplicial complex $K$ (i.e., the union of all vertices and edges). On the level of $k = 0$, observe that the inclusion of the 1-skeleton of $X$ induces a map of persistence modules of the form:

$$
\begin{array}{ccccccccc}
0 & \longrightarrow & H_0((Y_0)^1) & \longrightarrow & H_0((Y_1)^1) & \longrightarrow & ... & \longrightarrow & H_0((Y_m)^1) & \longrightarrow & H_0(\{*\}) \\
& & \downarrow & & \downarrow & & \downarrow & & \downarrow & & \parallel\downarrow \\
0 & \longrightarrow & H_0(Y_0) & \longrightarrow & H_0(Y_1) & \longrightarrow & ... & \longrightarrow & H_0(Y_m) & \longrightarrow & H_0(\{*\})
\end{array}
.$$

Here each vertical arrow is an isomorphism because the inclusion map $(Y_i)^1 \to Y_i$ is a bijection on connected components. Thus, we have an isomorphism of persistence modules and hence the top and bottom row have the same persistence diagrams. What this means is that 0-th dimensional persistent homology for the simplicial complex is equivalent to the 0-th dimensional persistent homology for its restriction to the 1-skeleton. Thus, this reduces to the scenario in Lemma 1, and this concludes the proof. □

# D    PROOFS FOR SECTION 6

*Proof of Theorem 2.* We split the proof of stability into two parts - the case where $k > 0$ and the case where $k = 0$.

Let $X$ be a simplicial complex and $H_k(X_\bullet, f, g)$ and $H_k(X_\bullet, f', g')$ be the two persistence modules in $(f, g)$-FB persistence and $(f', g')$-FB persistence respectively.

For $k > 0$, and when $X$ is a simplicial complex, we use the simplicial quotient interpretation of the persistence modules here (in the sense of Section 5). By Proposition 7, we see that the persistence pairs associated to $H_k(X_\bullet, f, g)$ is the same as the persistence pairs associated to the persistence homology of a simplicial filtrations of a simplicial complex $Z$, which we will write the filtration function as $h$. Similarly, for $H_k(X_\bullet, f', g')$, we will get a (possibly) different simplicial filtration of the same simplicial complex $Z$, which we will write the filtration function as $h'$.

Thus, we see that $d_B(H_k(X_\bullet, f, g), H_K(X_\bullet, f', g') = d_B(H_k(Z_\bullet, h), H_k(Z_\bullet, h'))$ is the bottleneck distance of the persistence diagrams coming from two filtration functions $h$ and $h'$. By the classic bottleneck stability of Cohen-Steiner et al. (2006), we have that

$$d_B(H_k(Z_\bullet, h), H_k(Z_\bullet, h')) \le ||h - h'||_\infty.$$

Let us unwrap the construction of $h$ and $h'$ here. Indeed, recall $Z$ is the simplicial cone of $X$. For $(f, g)$, the function $h$ agrees with $f$ when restricted to the base of the simplicial cone (which is a copy of $X$). Every other simplex must contain the disjoint basepoint $v_+$, and is obtained by joining $v_+$ to a base simplex $\sigma$ in $X$. We write all such simplices as $(\sigma, v_+)$. In this case, $h(\sigma, v_+)$ is defined to be $\max(f) + g(\sigma)$. Finally, one also specifies that $h(v_+)$ to be born before all the other persistence values, say 0 for uniformity (this does not affect the persistence pairs of dimension higher than 0). There is a similar description for $h'$, and hence we see that

$$\begin{aligned}
||h - h'||_\infty &= \max(||f - f'||_\infty, ||g - g' + (\max(f) - \max(f'))||_\infty) \\
&\le ||f - f'||_\infty + ||g - g' + (\max(f) - \max(f'))||_\infty \\
&\le ||f - f'||_\infty + ||g - g'||_\infty + |\max(f) - \max(f')|.
\end{aligned}$$

**Remark:** Note that there is no coefficient 2 here. For brevity, in the main paper, we added a 2 because that is the bound we will get for $k = 0$.

When $X$ is a (regular) cell complex, we evidently can still take the cone of a cell complex, which has cell decomposition according to the subcomplex of the base. The arguments above would still hold in this case. This is because the original Bottleneck stability Cohen-Steiner et al. (2006) was proven for all triangulable spaces and tame functions on them. Although not every cell complex is trianguable, it turns out every regular cell complex is trianguable (see Theorem 3.4.1 of Fritsch & Piccinini (1990)). The filtration is clearly still tame on the triangulation as the proof constructs the triangulation one skeleton at a time. It follows that the bound above applies through.

The case of $k = 0$ is different because the persistence tuples differ. First of all, by the same argument in the proof of Proposition 7, we can reduce this to the case where $X$ is a graph. We will still work in the simplicial quotient perspective. The reader might wonder - there are some graphs that are technically not simplicial complexes (i.e., has self-loops or multiple edges), does the simplicial quotient also work for them? The answer is yes because for such graphs, one can always discretize further by labeling the mid-point of exceptional edges (i.e., self-loops and multi-edges) as a vertex of the graph. This then becomes a simplicial complex and makes no difference in filtration since we will be spawning the midpoints at the same time as the edges, so we can ignore these trivial deaths.

Thus, we without loss reduce to the case where $X$ is a simplicial graph, and we can try to apply a variant of the simplicial quotient method. Indeed, we observe that, - if we move the adjoined formal basepoint $v_+$ to be spawned after we have filtrated the entire graph $X$ first using $f$ and before we start the contraction using $g$, then the 0-th dimensional persistence diagram with the pair representing $v_+$ removed is the 0-th dimensional persistence diagram from the 0-th dimensional $(f, g)$-persistence diagram. Indeed, this is because, $v_+$, being the last vertex spawned, will die immediately when the first vertex in $g$ appears and asks to be contracted (which will not change the other outputs).

To go further, we use a proof strategy of bottleneck stability (ex. Skraba & Turner (2021), Schnider (2024)) and adapt it to our scenario. Indeed, let $Z$ be the simplicial cone of $X$, and $h$ and $h'$ be the filtrations corresponding to the pairs $(f, g)$ and $(f', g')$. But also, because we are working with $k = 0$, we can again restrict to the 1-skeleton $Z'$ of $Z$, which is now a graph.

For each simplex (either an edge or a vertex) $x \in Z'$, we consider a linear interpolation function $h_t(y) = (1 - t)h(x) + th'(x)$ with $t$ values in $[0, 1]$. We can divide the interval $[0, 1]$ into a finite collection of intervals $[t_0, t_1], [t_1, t_2], ..., [t_n, t_{n+1}]$ with $t_0 = 0$ and $t_{n+1} = 1$ such that, for every pair of simplices $x$ and $y$ and $t \in [t_i, t_{i+1}]$, we either have that $h_t(x) \geq h_t(y)$ for all $t$ or $h_t(x) \leq h_t(y)$ for all $t$.

Since the relative ordering of simplices are not changed in this interval, the 0-dimensional persistence diagrams for $h_{t_i}$ and $h_{t_{i+1}}$ are the same in **combinatorial time**. Now choose any bijection $\pi$ of the **function time** persistence diagrams for $\mathrm{PH}_0^F(Z'; h_{t_i})$ and $\mathrm{PH}_0^F(Z'; h_{t_{i+1}})$ (by which we denote their forward time 0-diemnsional persistence diagrams) such that $\pi(b_t, d_t) = (b_{t+1}, d_{t+1})$ if and only if both tuples represent the same tuple in **combinatorial time**. Note that due to the placement of $v_+$, this will necessarily pair the two pairs associated to $v_+$ together (call them $(b_+, d_+)$ and $(b'_+, d'_+)$). Notably the restriction $\pi_1$ of $\pi$ to the tuples excluding the one corresponding to $v_+$ is still a bijection.

For any two filtrations $\phi, \psi$ of $Z'$ (the 1-skeleton of the cone of $X$). We write $d'_B(\mathrm{PH}_0^F(Z'; \phi), \mathrm{PH}_0^F(Z'; h_{t_{i+1}}))$ as the term

$$\inf_{\varphi \text{ bijections } \mathrm{PH}_0^F(Z';\phi) - (b_+, d_+) \to \mathrm{PH}_0^F(Z';\psi) - (b'_+, d'_+)} ||(b, d) - \varphi(b, d)||_\infty.$$

Observe that when $\phi = h$ and $\psi = h'$, one has that

$$d_B^{\mathrm{FB}}(H_0(X, f, g), H_0(X, f', g')) = d'_B(\mathrm{PH}_0^F(Z'; h), \mathrm{PH}_0^F(Z'; h')).$$

Now $d'_B$ can fail the non-degeneracy condition for being a metric, but it will still have a triangle inequality!

Now each tuple in the persistence diagram here actually comes from a **vertex-edge pair** $(v, e)$, where $v$ makes birth to the tuple and $e$ kills it. Choosing $\pi_1$ as the bijection here, we have

$$d'_B(\mathrm{PH}_0^F(Z'; h_{t_i}), \mathrm{PH}_0^F(Z'; h_{t_{i+1}})) \leq ||(b, d) - \pi_1(b, d)||$$
$$\leq \max_{(v,e) \in X} ||(h_{t_i}(v), h_{t_i}(e)) - (h_{t_{i+1}}(v), h_{t_{i+1}}(e))||_\infty$$
$$\text{Definition of } \pi_1 \text{ and associating pairs}$$
$$\leq \max_{v \in X} ||h_{t_i}(v) - h_{t_{i+1}}(v)||_\infty + \max_{e \in X} ||h_{t_i}(e) - h_{t_{i+1}}(e)||_\infty$$
$$\leq (t_{i+1} - t_i)||f - f'||_\infty + \max_{e \in X} ||h_{t_i}(e) - h_{t_{i+1}}(e)||_\infty$$

For the second term, there are two possibilities for the edge $e$ - either it comes from the filtration or it comes from the contraction. Thus, we have that

$$\max_{e \in X} ||h_{t_i}(e) - h_{t_{i+1}}(e)||_\infty \leq \max((t_{i+1} - t_i)||f - f'||_\infty,$$
$$(t_{i+1} - t_i)||(g - g') + (\max(f) - \max(f')||_\infty)$$
$$\leq (t_{i+1} - t_i)(||f - f'||_\infty + ||(g - g')||_\infty + |\max(f) - \max(f')|)$$

Now decomposing $d'_B(\mathrm{PH}_0^F(Z'; h), \mathrm{PH}_0^F(Z'; h'))$ into the sum $\sum_{i=0}^m d'_B(\mathrm{PH}_0^F(Z'; h_{t_i}), \mathrm{PH}_0^F(Z'; h_{t_{i+1}}))$ and using the bound above, we have that

$$d_B^{\mathrm{FB}}(H_0(X, f, g), H_0(X, f', g')) \leq 2||f - f'||_\infty + ||(g - g')||_\infty + |\max(f) - \max(f')|.$$

$\square$

## E    MORE DISCUSSIONS ON COMPARISON WITH OTHER METHODS

### E.1    ZIGZAG FILTRATION

In the definition of $(f, g)$-FB-persistence, the filtration function g is specifying the order of subgraphs being contracted. This is different from the use case of zigzag filtration Carlsson & de Silva (2010), where they are looking at a sequence of insertions and removals of a graph, but the removal process is fundamentally not continuous, so their persistence diagrams do not follow a linear time (whereas our set-up does).

To give an example, consider $G$ the path graph on 2 vertices, $H_1$ the emptyset, $H_2$ a subgraph of $G$ which is discrete with 2 vertices. Then the following is a diagram that can occur in zigzag filtration

$$H_1 \subset G \supset H_2$$

Observe that there is no map from $G$ to $H_2$ that is continuous, since we cannot split $G$ into 2 components. In general, the arrows in zigzag filtration go in different directions, and they are all inclusion maps. On the other hand, the class of diagrams we always consider in (f,g)-FB persistence is a sequence of continuous maps

$$G_1 \to G_2 \to G_3 \to ... \to G_n$$

which are inclusions followed by contractions. Here the arrows all go in the same direction, but they can be either inclusions or contractions (this is even more apparent in the setting of hourglass persistence).

More technically, zigzag filtration is considering quiver spaces on the path graph $P_n$ with any possible orientations but with inclusions, whereas we are considering quiver spaces on $P_n$ oriented in a uniform direction with both inclusions and contractions. Generally, we expect zigzag persistence and $(f, g)$-FB persistence to be incomparable, and their methods can complement each other.

### E.2    BIPATH FILTRATION

In Aoki et al. (2025), the authors introduced bipath persistence built on bipath filtrations, which are quiver spaces on the Hasse quiver $B_{n,m}$, which is of the form

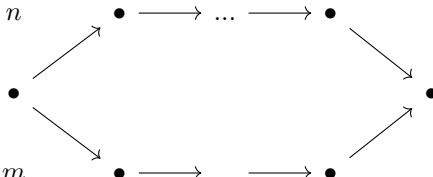

Similar to the case of zigzag filtration, the filtrations that appear in bipath filtration on Page 1 of Aoki et al. (2025) is again using inclusions only, whereas our settings consider the usage of contractions along with inclusions. Furthermore, bipath filtration considers quiver spaces on the Hasse quiver $B_{n,m}$ with inclusions, but we study quiver spaces on $P_n$ oriented in a uniform direction with both inclusions and contractions. Therefore, we expect that the two methods in general do not subsume one another but are complementary ideas.

### E.3    EFFICIENCY COMPARISON

For persistence diagrams (PDs) of dimension $> 0$ on simplicial complexes, hourglass persistence itself can be computed using the "simplicial quotient" trick mentioned in Section 5 of the paper using Proposition 7. Proposition 4 reduces the question to computing inclusion-based PH on a simplicial complex, which has a standard runtime dominated by the runtime to perform matrix reduction algorithms (see Otter et al. (2017)). For PDs of dimension 0, this can be reduced to looking at PDs of just underlying 1-skeleton (i.e., a graph). In this case, there is a way to optimize the calculation with union-find with a runtime dominated by $O(n \log n + m)$, where $n$ is the number

of edges and $m$ is the number of vertices (see Horn et al. (2021) for more details). Although Horn et al. (2021) only discuss the runtime in the filtration steps, tracking vertex representatives in the contraction steps would also have the same runtime.

When focusing on FB-persistence and $(f, g)-$FB-persistence, though, we expect the algorithms presented in Section 7 (which are not using the "simplicial quotient" trick) to be faster empirically. In general, the ability to reduce the size of object using contractions (ie. using the cellular complex extension in Section 5 instead) would lead to a decrease in the memory complexity. We therefore expect the cellular-based methods of hourglass persistence, $(f, g)-$FB persistence, etc. to have more efficient algorithms in the future.

### E.4 SCALING ON LARGE GRAPHS

Interleaving the filtration and contraction steps avoids the need to filtrate the entire graph before starting contractions, which allows the total size of the graph that appears in its entire lifetime to be bounded.

This can improve over the runtime on general simplicial complexes. Let $K$ be a simplicial complex with $n$ total simplicies. To give a heuristic/informal estimation from a practitioner perspective - the typical runtime of inclusion-based PH on $K$ scales approximately $O(n^3)$ as it requires a matrix reduction algorithm.

Suppose we bound the threshold to $d$ simplicies, and a practitioner equipped with the contraction framework can implement a specific instance of hourglass persistence as follows: as soon as we reach more than $d$ simplicies in a given step, we contract everything to a point, and so on. Then we believe the runtime should roughly to be $O(\frac{n}{d} \cdot (d)^3)$.

For practical purposes, it may also be beneficial to not contract everything to a point when the threshold is exceeded. If the contractions are done in multiple stages, then we believe a similar runtime analysis holds. We are excited at the possibilities that this framework can allow our community to address the problem of PH scaling on large graphs/simplicial complexes.

### E.5 MISCELLANEOUS DISCUSSIONS

Although we did not discuss this in the main paper, we remark that there is a suitable generalization of hourglass persistence to "$(f, g)$-hourglass persistence".

**Definition 15.** *Let $(f, g)$ be a pair of filtration functions on $G$, then the $(f, g)$-**hourglass persistence** is the persistence diagram of any sequence of inclusions of $\mathrm{IC}_i(G, f)$ and contractions of $\mathrm{IC}_j(G, g)$, provided that all the elements in $\mathrm{IC}_j(G, g)$ have appeared before it is being contracted.*

Unlike $(f, f)$-FB persistence, $(f, f)$-hourglass persistence would be strictly more expressive than forward persistence using $f$, due to the presence of permutations.

To conclude we also give one last remark about graph-pooling.

**Remark 4.** *Edge contractions have been used in graph pooling (Diehl et al., 2019; Limbeck et al., 2025) independently from persistent homology. We spectulate whether our approach can be applied with edge contractions in graph pooling.*

## F THE USE OF LARGE LANGUAGE MODELS

We used large language models to aid in the writing process of the introduction, to help check grammar / suggest edits for the main paper, and to double-check some algorithmic discussions arising from the paper.

