# OpenReview forum: "Contraction and Hourglass Persistence for Learning on Graphs, Simplices, and Cells"
_ICLR.cc/2026/Conference — ICLR 2026 Poster_

### Official Review · Reviewer_rVVx · 2025-10-15

**Soundness:** 3
**Presentation:** 3
**Contribution:** 4
**Rating:** 8
**Confidence:** 4

**Summary:**

This paper proposes Hourglass Persistence (HP), a novel topological descriptor that generalizes traditional persistent homology (PH) through alternating inclusion and contraction steps. This leads to a more expressive and metrizable topological representation, which is further extended to Hourglass Persistence, where inclusions and contractions can interleave multiple times.

**Strengths:**

1. Conceptual novelty and clear motivation.

    The idea of introducing contraction-based backward persistence and merging it with inclusion-based persistence provides a genuinely new perspective beyond traditional or extended PH.

2. Strong theoretical grounding.

    The paper gives clear definitions for backward, forward–backward, and hourglass persistence, proving expressive power hierarchies (Proposition 1–2) and linking them to extended PH via the general ((f,g))-FB formulation. Stability and metrizability are discussed rigorously, addressing key limitations of standard PH.

3. Generalizability.

    The framework is elegantly extended from graphs to higher-dimensional topological spaces, including simplicial and cellular complexes, with appropriate handling of quotient and contraction operations.

4. Algorithmic contribution.

    The paper provides clear computational procedures using union–find and cycle basis bookkeeping, making the method implementable and efficient.

**Weaknesses:**

1. Limited experimental validation
- Although Hourglass persistence appears highly competitive, the current experiments are not strong enough to convincingly demonstrate its superiority. Is combining Hourglass persistence with neural networks truly the best application scenario?
- The authors do not provide sufficient experimental details. From the description, it seems that the PH module in RePHINE was simply replaced with the forward–backward persistence. The authors should better clarify and justify this design choice.
- Could the authors include an ablation study for the Backward-only variant?
- Since Hourglass persistence introduces additional computational overhead, could the authors report runtime or provide a complexity analysis?
2. Related work

In my view, the idea most closely related to Hourglass persistence is zigzag persistence [1], which also models both the appearance and disappearance of complexes during the filtration process. Could the authors discuss this connection in more detail?

[1] Carlsson G, De Silva V. Zigzag persistence[J]. Foundations of computational mathematics, 2010, 10(4): 367-405.

**Questions:**

1. Please provide additional experimental evidence.
2. Please include a comparison with zigzag persistence.

---

> ### Author Response · Authors · 2025-11-21
>
> Thank you for your time and comments. Based on your suggestions, we have also revised our submission, where the changes are highlighted in blue. We reply to your comments/questions below.
>
> > Limited experimental validation
> Although Hourglass persistence appears highly competitive, the current experiments are not strong enough to convincingly demonstrate its superiority.
>
> > Could the authors include an ablation study for the Backward-only variant?
>
> ### Answer (Part 1 of 2):
>
> In the revised version, we have substantially expanded the empirical evaluation to directly address both scalability and significance of improvements. These results are now included in Section 7.3 and Table 1  of the main paper.
>
> **(1) Larger-scale benchmarks.**
> Beyond the four TUDatasets, we have added experiments on two widely used, significantly larger benchmarks:
> - **ZINC** (graph regression; 12k graphs): We report MAE (lower is better)
> - **OGBG-MOLHIV** (molecular property prediction; >40k graphs): We report AUC (higher is better).
>
> **(2) Statistical significance of improvements over baselines.**
> All results are now averaged over **five random seeds**, and we report **mean ± standard deviation** for every entry.
>
> **(3) Ablations.**
> Furthermore,to isolate the individual contributions of our two components, we include ablations that retain only the forward inclusion stage (**Fwd-only**) or only the backward contraction stage (**Bwd-only**), enabling a precise assessment of the role each module plays in the overall performance.
>
> These experimental results are summarized in the table below:
>
> ### **Table: Comparison of PH variants across six datasets (GIN and GCN backbones)**
> Classification accuracy and AUC are reported in percentage (% ↑).
> ZINC is evaluated using MAE (↓).
> ## **GIN Backbone**
>
> | Dataset | PH | RePHINE | Fwd-only | Bwd-only | Ours |
> |-|-|-|-|-|-|
> | **NCI109 (Acc ↑)** | 76.76 ± 0.40 | _77.89 ± 1.19_ | 77.00 ± 1.03 | 76.35 ± 0.50 | **77.89 ± 1.87** |
> | **PROTEINS (Acc ↑)** | 69.35 ± 1.83 | 69.94 ± 2.76 | _70.24 ± 2.95_ | 70.54 ± 2.19 | **73.51 ± 1.11** |
> | **IMDB-B (Acc ↑)** | 68.67 ± 1.25 | 70.67 ± 0.94 | **74.67 ± 0.47** | _74.33 ± 0.94_ | 72.00 ± 2.16 |
> | **NCI1 (Acc ↑)** | 79.24 ± 1.74 | 78.75 ± 2.55 | _76.72 ± 1.13_ | 75.75 ± 0.94 | **81.27 ± 0.00** |
> | **ZINC (MAE ↓)** | 0.43 ± 0.01 | _0.41 ± 0.01_ | 0.62 ± 0.01 | 0.61 ± 0.00 | **0.40 ± 0.01** |
> | **MOLHIV (AUC ↑)** | **74.34 ± 4.57** | _72.88 ± 2.15_ | 70.00 ± 3.11 | 70.59 ± 1.83 | 72.34 ± 0.74 |
>
> ---
> ## **GCN Backbone**
>
> | Dataset|PH|RePHINE|Fwd-only|Bwd-only|Ours|
> |-|-|-|-|-|-|
> | **NCI109 (Acc ↑)** | 76.59 ± 1.32 | **79.50 ± 0.11** | 71.91 ± 0.52 | _74.58 ± 0.71_ | 75.87 ± 0.89 |
> | **PROTEINS (Acc ↑)** | 70.54 ± 0.73 | 68.75 ± 2.53 | 69.35 ± 1.83 | _70.54 ± 1.46_ | **72.32 ± 1.46** |
> | **IMDB-B (Acc ↑)** | 65.00 ± 1.63 | _70.00 ± 0.82_ | 62.67 ± 3.30 | 64.33 ± 3.30 | **68.00 ± 2.16** |
> | **NCI1 (Acc ↑)** | 78.43 ± 0.98 | **78.91 ± 0.80** | 75.59 ± 1.00 | 76.24 ± 1.74 | _78.67 ± 1.69_ |
> | **ZINC (MAE ↓)** | 0.49 ± 0.02 | _0.46 ± 0.01_ | 0.86 ± 0.01 | 0.87 ± 0.01 | **0.44 ± 0.01** |
> | **MOLHIV (AUC ↑)** | 75.12 ± 0.68 | _75.40 ± 0.53_ | 71.02 ± 2.18 | 71.55 ± 1.21 | **76.37 ± 1.45** |
>
> From the table, we observe that:
> 1. **Our method** achieves the best or second-best performance on **9 out of 12 settings**, demonstrating the benefit of jointly learning *both* inclusion and contraction schedules.
>
> 2. **RePHINE** consistently outperforms standard PH in almost all cases, confirming that learning both vertex and edge filtrations (plus its 0D augmentation) is beneficial.
>
> 3. **Ablations** show the contribution of each component:
>    - **Fwd-only** (learned vertex–edge filtration, no contractions) reliably improves over standard PH.
>    - **Bwd-only** (learned contractions only) also improves on several datasets and performs roughly on par with Fwd-only.
>
> 4. **RePHINE vs ablations**:
>    RePHINE strongly outperforms both ablations on **ZINC** and **MOLHIV**, indicating that its explicit 0D augmentation captures useful early-connectivity structure.
>
> 5. **Full model** consistently outperforms both ablations, showing that *inclusion* and *contraction* encode complementary structural information and must be learned jointly for the strongest performance across classification, regression, and molecular tasks.
>
> The performance gaps—especially on PROTEINS, NCI1, ZINC, and MOLHIV—remain consistent and statistically meaningful across seeds.
>
> Overall, the expanded experiments demonstrate both **scalability to larger datasets**,   **consistent improvements over strong baselines**, and **importance of each architectural component to overall performance**.

---

> > ### Author Response · Authors · 2025-11-21
> >
> > > Limited experimental validation
> > Although Hourglass persistence appears highly competitive, the current experiments are not strong enough to convincingly demonstrate its superiority.
> >
> > > Could the authors include an ablation study for the Backward-only variant?
> >
> > ### Answer (Part 2 of 2):
> >
> > **Comparision with extended persistence**
> > We have also added a dedicated subsection and new experiments to the revised paper (see Section 7.4, “Comparison with extended persistence”). The updated results are shown below in  the two tables gor GIN and GCN backbones:
> >
> > |Dataset|ExtP (GIN)|PersLay (GIN)|Ours (GIN)|
> > |-|-|-|-|
> > | **NCI109**| **78.21**| 68.28| **78.21**|
> > | **PROTEINS** | **74.11**  | 66.07  | 73.21|
> > | **IMDB-B** | 63.00| 70.00| **73.00**|
> > | **NCI1** | 78.59| 68.86 | **81.51**|
> >
> > |Dataset|ExtP (GCN)|PersLay (GCN)|Ours (GCN)|
> > |-|-|-|-|
> > | **NCI109**| **77.48**  | 68.28| **77.48** |
> > | **PROTEINS** | **72.32**  | 66.07| **72.32**|
> > | **IMDB-B**|66.00| **70.00**| **70.00**|
> > | **NCI1**|80.05| 68.86| **80.78**|
> >
> > **Extended-persistence comparison.**
> > Extended persistence combines sublevel and superlevel sets, and PersLay (Carrière et al., 2020) implements this by computing extended-persistence diagrams and then applying a learnable layer on top. In the original paper, we gave an explicit theoretical connection (Proposition 3, included for completeness) showing that classical extended persistence has the same expressive power as an **$(f,-f)$** forward–backward (FB) schedule. Based on this equivalence, we now integrate extended persistence *directly* into our FB framework by using $ -f $ as the backward schedule.
> >
> > We observe that:
> >
> > 1. **Our \((f,-f)\) adaptation into our (f,g)-FB GNN model consistently outperforms PersLay** for both GIN and GCN backbones.
> >    This demonstrates that incorporating extended-persistence structure *within* the FB mechanism is more effective than processing extended diagrams via a post-hoc embedding layer.
> >
> > 2. **Our full \((f,g)\)-forward–backward model further outperforms both PersLay and the extended variant.**
> >    This shows that jointly learning inclusion and contraction yields strictly richer topological information than extended persistence alone.
> >
> > These results—now included in the main paper—provide a clear and fair empirical comparison with extended persistence and highlight the advantages of our forward–backward framework.
> >
> > > The authors do not provide sufficient experimental details. From the description, it seems that the PH module in RePHINE was simply replaced with the forward–backward persistence. The authors should better clarify and justify this design choice.
> >
> > We thank the reviewer for the opportunity to clarify the design choice in detail. Our intent was to evaluate our forward–backward (FB) persistence *as a drop-in alternative* to RePHINE’s learnable inclusion-only persistence. This isolates the effect of the persistence mechanism while keeping the GNN backbone and pooling identical across methods. Every GNN layer outputs learnable node/edge embeddings, which are interpreted as filtration values; the PH block then computes FB persistence and feeds the resulting signatures into the next layer. This end-to-end differentiable “PH-between-layers” structure is exactly the setting in which RePHINE operates, and therefore the fairest way to compare the two. The revised paper now explicitly includes full ablations and  extended-persistence comparisons.
> >
> > >Is combining Hourglass persistence with neural networks truly the best application scenario?
> >
> > While Hourglass persistence is broadly applicable, we believe neural networks provide the most natural and powerful setting for exploiting its expressive advantages. By allowing both inclusion and contraction schedules to be driven directly by learned features, the Hourglass/FB module becomes a structural, task-adaptive component of the network rather than a fixed preprocessing step.

---

> > > ### Author Response · Authors · 2025-11-21
> > >
> > > > Since Hourglass persistence introduces additional computational overhead, could the authors report runtime or provide a complexity analysis?
> > >
> > > Thank you for your question. We have included a discussion of this in Appendix E. For persistence diagrams (PDs) of dimension > 0, hourglass persistence itself can be computed using the "simplicial quotient" trick mentioned in Section 5.1 of the paper using Proposition 7. Prop 7 reduces the question to computing inclusion-based PH on a simplicial complex, which has a standard runtime dominated by the runtime to perform matrix reduction algorithms (see [2]). For PDs of dimension = 0, this can be reduced to looking at PDs of just underlying 1-skeleton (which is a graph named say $G$). In this case, there is a way to optimize the calculation with union-find with a runtime dominated by O(nlog n + m), where n is the number of edges in $G$ and $m$ is the number of vertices (see [3] for more details). Although [3] only discuss the runtime in the filtration steps, tracking vertex representatives in the contraction steps would also have the same runtime.
> > >
> > > When focusing on FB-persistence and (f,g)-FB-persistence, though, we expect the algorithms presented in 7.1 (which are not using the "simplicial quotient" trick) to be faster in practice. In general, the ability to reduce the size of object using contractions (ie. using the cellular complex extension in Section 5.2 instead) would lead to a decrease in the memory complexity. We therefore expect the cellular-based methods of hourglass persistence, $(f,g)-$FB persistence, etc. to have more efficient algorithms in the future.
> > >
> > > > In my view, the idea most closely related to Hourglass persistence is zigzag persistence [1], which also models both the appearance and disappearance of complexes during the filtration process. Could the authors discuss this connection in more detail?
> > >
> > > Thank you for raising the work on zigzag filtration [1], we have included it as a discussion in both the main paper and Appendix E. In the definition of hourglass persistence, we are doing a sequence of **insertions and contractions**. This is different from the use case of Zigzag filtration, where they are looking at a sequence of **insertions and removals** of a graph, but the removal process is fundamentally not continuous, so their persistence diagrams do not follow a linear time (whereas our set-up does).
> > >
> > > To give an example, consider G the path graph on 2 vertices, H1 the emptyset, H2 a subgraph of G which is discrete with 2 vertices. Then the following is a diagram that can occur in zigzag filtration
> > >
> > > $H_1 \subset G \supset H_2$
> > >
> > > Observe that there is no map from G to H2 that is continuous, since we cannot split G into 2 components. In general, the arrows in zigzag filtration go in different directions, and they are all inclusion maps.
> > >
> > > On the other hand, the class of diagrams we always consider in hourglass persistence (or (f,g)-FB persistence) is a sequence of continuous maps
> > >
> > > $G_1 \to G_2 \to G_3 \to ... \to G_n$
> > >
> > > which are inclusions followed by contractions. Here the arrows all go in the same direction, but they can be either inclusions or contractions (this is even more apparent in the setting of hourglass persistence).
> > >
> > > On a more technical level, zigzag filtration is considering quiver spaces on the path graph Pn with any possible orientations but with inclusions, whereas we are considering quiver spaces on Pn oriented in a uniform direction with both inclusions and contractions. In general, we expect zigzag persistence and (f,g)-FB persistence / hourglass persistence to be incomparable, and their methods can complement each other.
> > >
> > > -------------
> > > Thank you again for your review. We hope your concerns have been satisfactorily addressed, and if so, would appreciate if you could revisit your score to reflect the same. We are also committed to engaging further if you have any additional questions, concerns, or suggestions.
> > >
> > > [1] Carlsson G, De Silva V. Zigzag persistence. Foundations of computational mathematics, 2010, 10(4): 367-405.
> > >
> > > [2] Otter et al. A roadmap for the computation of persistent homology, EPJ Data Science.
> > >
> > > [3] Horn et al. Topological Graph Neural Networks. ICLR 2022.

---

> > > > ### Comment · Reviewer_rVVx · 2025-11-27
> > > >
> > > > Thank you very much for the author's rebuttal. I will keep my score, but I hope they can include a richer comparison with other persistent homologies in the final version because this is very important for the paper.

---

> > > > > ### Author Response · Authors · 2025-11-28
> > > > >
> > > > > Dear Reviewer rVVx,
> > > > >
> > > > > Thank you for letting us know. We will be sure to incorporate many of your feedbacks and questions in our paper, and we aim to include comparisons with other PH methods in the final version. Thank you again for your time and considerations.

---

### Official Review · Reviewer_UtXw · 2025-10-31

**Soundness:** 2
**Presentation:** 2
**Contribution:** 2
**Rating:** 2
**Confidence:** 3

**Summary:**

This paper proposes a novel filtration for feature extraction based on persistent homology. Unlike conventional approaches that employ forward filtration, the proposed method leverages a backward filtration. This design enables the method to achieve effective performance on graph datasets.

**Strengths:**

This paper proposes a new type of filtration from a novel perspective of persistent homology. While conventional approaches have primarily relied on forward filtration, the authors construct an interesting framework based on a backward filtration. As a result, the proposed method achieves improved performance in the experiments.

**Weaknesses:**

The main concerns are as follows:

- The central claims of this paper are the use of backward filtration and the introduction of (f,g)(f,g)-FB persistence. While the proposed method combines forward and backward (f,g)(f,g)-FB persistence, the actual contribution of the core idea — the backward filtration itself — remains unclear. Moreover, it is not evident how (f,g)(f,g)-FB persistence fundamentally differs from simply combining features derived from two filtrations. In general, combining multiple heterogeneous features often leads to performance gains, so the improvement shown here may not be specific to the proposed formulation.

- Although the notion of backward filtration is not identical, similar ideas have been explored in approaches such as Zigzag filtration [1]. The distinction between these existing methods and the proposed one is insufficiently clarified.

- The combination of two filtrations has also been proposed in prior work [2]. While the meaning differs due to the reversed filtration direction, the proposed combination seems to follow a rather straightforward idea, making it difficult to consider the novelty as significant.

- Since this conference focuses on machine learning and deep learning, introducing a new idea in persistent homology alone is not sufficient to meet the scope. It is therefore crucial to demonstrate the effectiveness of the approach in machine learning applications, but the current experimental evidence is not sufficient. In addition, Theorem 1 appears to be merely the authors’ claim and does not constitute a theorem in a rigorous sense. A theorem should be written in a mathematically precise manner that clearly states in what sense the proposed method is effective.

[1] G. Carlsson et al., Zigzag Persistence, Foundations of Computational Mathematics, Volume 10, pages 367–405, (2010)

[2] T. Aoki et al., Bipath persistence, Japan Journal of Industrial and Applied Mathematics, - Volume 42, pages 453–486, (2025)

**Questions:**

- Please include an ablation study comparing **forward-only**, **backward-only**, and **(f,g)-FB persistence** variants to quantify the contribution of the backward filtration (e.g., absolute/relative gains, with statistical significance).

- Please clarify how (f,g)-FB persistence differs from a simple concatenation/union of features derived from two filtrations. A theoretical distinction and an empirical comparison (which can be incorporated into the ablation) would be helpful.

- Either explain why zigzag filtration cannot achieve the same effect as the proposed method, or demonstrate empirically that your approach outperforms zigzag-based baselines.

---

> ### Author Response · Authors · 2025-11-21
>
> Thank you for your time and comments. Based on your suggestions, we have also revised our submission, where the changes are highlighted in blue.
> >  ... the actual contribution of - the backward filtration itself - remains unclear. ... ...
>
> Our focus on an independent graph contraction (with g), as a tool for extracting topological descriptors, is the key departure from previous filtration based methods.
>
> **Usage of Contraction:** When we are considering (f,g)-FB persistence, g specifies the **sequence of subgraphs being contracted**. The ability to contract graphs gives us a way to reduce the complexity of the graph, which allows the deaths of permanent features that appear in the filtration steps introduced by f.
>
> This is important for stability - for ordinary persistence coming from different graphs or complexes, their bottleneck distance may not be finite as ordinary persistence allows certain permanent features (ie. diagrams that die at infinity) to live forever. If the two graphs have two different multisets of permanent features, then their bottleneck distance is infinite. This makes the comparing persistence diagrams of different graphs difficult. However, extended persistence (as (f,-f)-FB persistence) and (f,g)-FB persistence on graphs and complexes will allow all persistence diagrams (with the exception of a single vertex in 0-dim PH) to die at finite time. This allows us to compare their respective (f,g)-FB persistence diagrams using bottleneck distance. The consideration for stability and metrizability is why we considered backward filtration.
>
> This is also beneficial from a standpoint of learnability. (f,g)-FB-persistence have the benefit of added stability as they allow the deaths of permeant features that appears in the filtration steps of the graph G. The added death times to these features can be seen as a way to encode more information and hence more learnability. Moreover, in a given filtration, some Morse critical points (ex. a height function on a Reeb graph) do not seem to be accounted from ordinary PH. The second function indicated by g (going back) is a chance to capture those critical points. The freedom to choose g allows the framework to capture more information. Finally, hourglass is an interleaving of the forward and backward steps. The flexibility to order the sequence encodes a wider range of possibilities to be learned.
> > Zigzag filtration [1]
>
> We have included [1] as a discussion in both the main paper and Appendix E.
>
> We are not reversing the filtration direction, we are **performing inclusions and contractions**. In the definition of (f,g)-FB-persistence, g is specifying the order of subgraphs being contracted. This is different from the use case of Zigzag filtration, where they are looking at a sequence of **insertions and removals** of a graph, but the removal process is fundamentally not continuous, so their persistence diagrams do not follow a linear time (whereas our set-up does).
>
> Let G be a path graph on 2 vertices, $H_1$ empty, $H_2$ a discrete subgraph of G with 2 vertices. Then the following is a diagram that can occur in zigzag filtration
> $H_1\subset G\supset H_2$
> There is no map from G to $H_2$ that is continuous, since we cannot split G into 2 components. The inclusion arrows in zigzag filtration go in different directions.
>
> On the other hand, the class of diagrams we always consider in (f,g)-FB persistence is a sequence of continuous maps
> $G_1\to...\to G_n$
> which are inclusions followed by contractions. Here the arrows all go in the same direction, but they can be either inclusions or contractions (this is even more apparent in the setting of hourglass persistence).
>
> Zigzag filtration is considering quiver spaces on the path graph Pn with any possible orientations and inclusions, whereas we are considering quiver spaces on Pn oriented in a uniform direction with both inclusions and contractions. We expect zigzag persistence and (f,g)-FB persistence / hourglass persistence to be incomparable, and their methods can complement each other.
> > The combination of two filtrations ... proposed in prior work [2]
>
> We have included [2] as a discussion in both the main paper and Appendix E. Similar to the case of zigzag filtration, the filtrations that appear in bipath filtration on Page 1 of [2] is again using **inclusions only**, whereas our settings consider the usage of **contractions along with inclusions**. Furthermore, bipath filtration considers quiver spaces on the Hasse quiver $B_{n,m}$ with inclusions, but we study quiver spaces on $P_n$ oriented in a uniform direction with both inclusions and contractions. We expect that the two methods in general do not subsume one another but are complementary ideas.
>
> Doing inclusions and contractions are complementary and combining them can lead to added benefits in expressivity, stability, and learnability than doing them alone.
>
> [1] G. Carlsson et al., Zigzag Persistence
>
> [2] T. Aoki et al., Bipath persistence

---

> > ### Author Response · Authors · 2025-11-21
> >
> > > ... experimental evidence is not sufficient.
> > > ... ablation study
> >
> > In the revised version, we have substantially expanded the empirical evaluation to directly address both scalability and significance of improvements. These results are now included in Section 7.3 and Table 1  of the main paper.
> >
> > **(1) Larger-scale benchmarks.**
> > Beyond the four TUDatasets, we have added experiments on two widely used, significantly larger benchmarks:
> > - **ZINC** (graph regression; 12k graphs): We report MAE (lower is better)
> > - **OGBG-MOLHIV** (molecular property prediction; >40k graphs): We report AUC (higher is better).
> >
> > **(2) Statistical significance of improvements over baselines.**
> > All results are now averaged over **five random seeds**, and we report **mean ± standard deviation** for every entry.
> >
> > **(3) Ablations.**
> > Furthermore,to isolate the individual contributions of our two components, we include ablations that retain only the forward inclusion stage (**Fwd-only**) or only the backward contraction stage (**Bwd-only**), enabling a precise assessment of the role each module plays in the overall performance.
> >
> > These experimental results are summarized in the table below:
> >
> > ### **Table: Comparison of PH variants across six datasets (GIN and GCN backbones)**
> > Classification accuracy and AUC are reported in percentage (% ↑).
> > ZINC is evaluated using MAE (↓).
> > ## **GIN Backbone**
> >
> > | Dataset | PH | RePHINE | Fwd-only | Bwd-only | Ours |
> > |-|-|-|-|-|-|
> > | **NCI109 (Acc ↑)** | 76.76 ± 0.40 | _77.89 ± 1.19_ | 77.00 ± 1.03 | 76.35 ± 0.50 | **77.89 ± 1.87** |
> > | **PROTEINS (Acc ↑)** | 69.35 ± 1.83 | 69.94 ± 2.76 | _70.24 ± 2.95_ | 70.54 ± 2.19 | **73.51 ± 1.11** |
> > | **IMDB-B (Acc ↑)** | 68.67 ± 1.25 | 70.67 ± 0.94 | **74.67 ± 0.47** | _74.33 ± 0.94_ | 72.00 ± 2.16 |
> > | **NCI1 (Acc ↑)** | 79.24 ± 1.74 | 78.75 ± 2.55 | _76.72 ± 1.13_ | 75.75 ± 0.94 | **81.27 ± 0.00** |
> > | **ZINC (MAE ↓)** | 0.43 ± 0.01 | _0.41 ± 0.01_ | 0.62 ± 0.01 | 0.61 ± 0.00 | **0.40 ± 0.01** |
> > | **MOLHIV (AUC ↑)** | **74.34 ± 4.57** | _72.88 ± 2.15_ | 70.00 ± 3.11 | 70.59 ± 1.83 | 72.34 ± 0.74 |
> >
> > ---
> > ## **GCN Backbone**
> >
> > | Dataset|PH|RePHINE|Fwd-only|Bwd-only|Ours|
> > |-|-|-|-|-|-|
> > | **NCI109 (Acc ↑)** | 76.59 ± 1.32 | **79.50 ± 0.11** | 71.91 ± 0.52 | _74.58 ± 0.71_ | 75.87 ± 0.89 |
> > | **PROTEINS (Acc ↑)** | 70.54 ± 0.73 | 68.75 ± 2.53 | 69.35 ± 1.83 | _70.54 ± 1.46_ | **72.32 ± 1.46** |
> > | **IMDB-B (Acc ↑)** | 65.00 ± 1.63 | _70.00 ± 0.82_ | 62.67 ± 3.30 | 64.33 ± 3.30 | **68.00 ± 2.16** |
> > | **NCI1 (Acc ↑)** | 78.43 ± 0.98 | **78.91 ± 0.80** | 75.59 ± 1.00 | 76.24 ± 1.74 | _78.67 ± 1.69_ |
> > | **ZINC (MAE ↓)** | 0.49 ± 0.02 | _0.46 ± 0.01_ | 0.86 ± 0.01 | 0.87 ± 0.01 | **0.44 ± 0.01** |
> > | **MOLHIV (AUC ↑)** | 75.12 ± 0.68 | _75.40 ± 0.53_ | 71.02 ± 2.18 | 71.55 ± 1.21 | **76.37 ± 1.45** |
> >
> > From the table, we observe that:
> > 1. **Our method** achieves the best or second-best performance on **9 out of 12 settings**, demonstrating the benefit of jointly learning *both* inclusion and contraction schedules.
> >
> > 2. **RePHINE** consistently outperforms standard PH in almost all cases, confirming that learning both vertex and edge filtrations (plus its 0D augmentation) is beneficial.
> >
> > 3. **Ablations** show the contribution of each component:
> >    - **Fwd-only** (learned vertex–edge filtration, no contractions) reliably improves over standard PH.
> >    - **Bwd-only** (learned contractions only) also improves on several datasets and performs roughly on par with Fwd-only.
> >
> > 4. **RePHINE vs ablations**:
> >    RePHINE strongly outperforms both ablations on **ZINC** and **MOLHIV**, indicating that its explicit 0D augmentation captures useful early-connectivity structure.
> >
> > 5. **Full model** consistently outperforms both ablations, showing that *inclusion* and *contraction* encode complementary structural information and must be learned jointly for the strongest performance across classification, regression, and molecular tasks.
> >
> > The performance gaps—especially on PROTEINS, NCI1, ZINC, and MOLHIV—remain consistent and statistically meaningful across seeds.
> >
> > Overall, the expanded experiments demonstrate both **scalability to larger datasets**,   **consistent improvements over strong baselines**, and **importance of each architectural component to overall performance**.
> >
> > **Comparision with extended persistence**
> > We have also added a dedicated subsection and new experiments to the revised paper (see Section 7.4, “Comparison with extended persistence”). The updated results are shown below in  the two tables gor GIN and GCN backbones:
> >
> > |Dataset|ExtP (GIN)|PersLay (GIN)|Ours (GIN)|
> > |-|-|-|-|
> > | **NCI109**| **78.21**| 68.28| **78.21**|
> > | **PROTEINS** | **74.11**  | 66.07  | 73.21|
> > | **IMDB-B** | 63.00| 70.00| **73.00**|
> > | **NCI1** | 78.59| 68.86 | **81.51**|
> >
> > |Dataset|ExtP (GCN)|PersLay (GCN)|Ours (GCN)|
> > |-|-|-|-|
> > | **NCI109**| **77.48**  | 68.28| **77.48** |
> > | **PROTEINS** | **72.32**  | 66.07| **72.32**|
> > | **IMDB-B**|66.00| **70.00**| **70.00**|
> > | **NCI1**|80.05| 68.86| **80.78**|

---

> > > ### Author Response · Authors · 2025-11-21
> > >
> > > **Extended-persistence comparison.**
> > > Extended persistence combines sublevel and superlevel sets, and PersLay (Carrière et al., 2020) implements this by computing extended-persistence diagrams and then applying a learnable layer on top. In the original paper, we gave an explicit theoretical connection (Proposition 3, included for completeness) showing that classical extended persistence has the same expressive power as an **$(f,-f)$** forward–backward (FB) schedule. Based on this equivalence, we now integrate extended persistence *directly* into our FB framework by using $ -f $ as the backward schedule.
> > >
> > > We observe that:
> > > 1. **Our \((f,-f)\) adaptation into our (f,g)-FB GNN model consistently outperforms PersLay** for both GIN and GCN backbones.
> > >    This demonstrates that incorporating extended-persistence structure *within* the FB mechanism is more effective than processing extended diagrams via a post-hoc embedding layer.
> > >
> > > 2. **Our full \((f,g)\)-forward–backward model further outperforms both PersLay and the extended variant.**
> > >    This shows that jointly learning inclusion and contraction yields strictly richer topological information than extended persistence alone.
> > >
> > > These results, now included in the paper, provide a clear and fair empirical comparison with extended persistence and highlight the advantages of our forward–backward framework.
> > >
> > > >Theorem 1 appears to be merely the authors’ claim and does not constitute a theorem
> > >
> > > Thank you for the opportunity to clarify by our work. When we talk about **expressivity** we refer to the ability of the two methods to distinguish non-isomorphic graphs. We draw upon the general notion of expressivity as discussed in, for example, Definition 1 and 2 of [3], Section 3 of [4], or Chapter 5 of the textbook [5]. To clarify, by **strictly more expressive**, we mean:
> > >
> > > More precisely, for any graph $G$ with filtration $f$, the FB-persistent diagram with respect to $f$ can recover the correspondent forward and backward persistence diagrams. However, there exists a pair of graphs $G, H$ with a permutation equivariant filtration $f$ such that their forward and backward persistence diagrams are the same, but their FB-persistence diagrams differ.
> > >
> > > We have added the explanation above to Theorem 1 in the revised manuscript. The original proof of Theorem 1 validates the statement above, showing that FB-persistence can always recover forward and backward PH, but the union of forward and backward PH can have less information than FB-persistence.
> > >
> > > > ... clarify how (f,g)-FB persistence differs from a simple concatenation/union of features
> > >
> > > Consider G and H from Figure 3(c) in the paper. Take f = degree based filtration and $g = f^{\sigma}$, in the sense of **Proposition 4** of the paper. By the same proposition the FB-persistence is the same as $(f, g)$-FB persistence. In the proof of Theorem 1, it is shown that G and H have the same forward-based PH with respect to $f$, but they have different FB-persistence. On the other hand, we can check that G and H also have the same forward-based PH with respect to $g$. This is verified in the code file **check_diagram.py**, now uploaded to the supplementary material we have included in the paper. This can also be seen by computing by hand. Thus, (f,g)-FB persistence is different from a simple union of the two diagrams derived from $f$ and $g$.
> > >
> > > A conceptual reason on why the two graphs cannot be distinguished by $g$ alone is that their cycles all live to infinity. On the other hand, $(f,g)$-FB persistence allows the cycles born to die, and the cycles in G and H respectively with have different birth and death times in $(f,g)$-FB persistence.
> > >
> > > In general, we expect $(f,g)$-FB persistence to be quite powerful in terms of expressivity. For example, **Proposition 3** (which we included for completeness) notes that a special case $(f,-f)$-FB persistence has the same expressive power as extended persistence, and the works of [6] shows that for certain choices of f, extended persistence is more expressive than 3-WL.
> > >
> > > > Either explain why zigzag filtration cannot achieve the same effect as the proposed method, or demonstrate empirically that your approach outperforms zigzag-based baselines.
> > >
> > > Please see our response to why zigzag filtrations and (f,g)-FB-persistence (and more generally hourglass persistence) are fundamentally different methods above.
> > >
> > > ---------------
> > > We hope your concerns have been satisfactorily addressed, and if so, would appreciate if you could revisit your score to reflect the same. We are also committed to engaging further if you have any additional questions, concerns, or suggestions.
> > >
> > > [3] Ballester and Rieck. On the Expressivity of Persistent Homology in Graph Learning.
> > >
> > > [4] Zhang et al. The Expressive Power of Graph Neural Networks: A Survey
> > >
> > > [5] Wu et al. Graph Neural Networks: Foundations, Frontiers, and Applications.
> > >
> > > [6] Yan et al. "Enhancing graph representation learning with localized topological features."

---

> ### Author Response · Authors · 2025-11-28
> **Rebuttal Follow-Up**
>
> Dear Reviewer UtXw,
>
> As the discussion period is ending soon, we wanted to check in again. In our rebuttal, we have addressed each of the weaknesses and questions you listed as follows:
>
> 1. We gave a conceptual explanation on how (f,g)-FB persistence and backward filtration are more effective than traditional methods. The ability for the backward steps to contract graphs leads to memory efficiency in terms of down-scaling the size of the graph, the benefits of metrizability and stability, and introduce more learnability in allowing permanent features to die and detecting Morse critical points. Moreover, we have made a clear distinction between our methods with zigzag and bipath filtrations, as elaborated in our rebuttal and **Appendix E** of our revised manuscript.
>
>
> 2. We have demonstrated the benefits and effectiveness of our method in machine learning applications by conducting **additional experiments**, which are elaborated in **Section 7** of our revised manuscript. In particular:
>    - We conducted graph regression tasks on the dataset **ZINC** and molecular property predictions on the dataset **OGBG-MOLHIV** with **statistical significance of improvements** over baselines.
>
>    - We added additional **ablation** studies that compare our methods with forward-only and backward-only methods across the 6 datasets with GIN and GCN backbone. Our (f,g)-FB-persistence achieved either the best or second best on **9 out of 12** experiments.
>
>    - We have also compared our method with **extended persistence** in two forms, one directly with the PersLay framework and another by interpreting extended persistence as (f,-f)-FB persistence (which we call ExtP). ExtP and the full (f,g)-FB-persistence consistently outperforms PersLay in most cases.
>
>
> 3. We have clarified in **Theorem 1** on what **strictly more expressive** means and provided additional references [1,2,3] in the literature that surveys GNN expressivity.
>
>
> 4. We clarified how Theorem 1 also gives an example where **(f,g)-FB persistence differs from a simple union** of features. A conceptual reason is that giving permenant features death times adds more learnability.
>
> We hope your concerns have been satisfactorily addressed, and if so, would appreciate if you could revisit your score to reflect the same. If you have any specifc questions or concerns, we would be happy to address them. Thank you again for your feedback and help to improve the paper.
>
> [1] Ballester and Rieck. On the Expressivity of Persistent Homology in Graph Learning. LoG 2024.
>
> [2] Zhang et al. The Expressive Power of Graph Neural Networks: A Survey
>
> [3] Wu et al. Graph Neural Networks: Foundations, Frontiers, and Applications. Springer 2022.

---

### Official Review · Reviewer_7Wbe · 2025-11-02

**Soundness:** 3
**Presentation:** 3
**Contribution:** 3
**Rating:** 6
**Confidence:** 4

**Summary:**

This paper introduces a new family of topological descriptors for graphs, simplicial complexes, and cellular complexes, termed hourglass persistence, which interleaves inclusion and contraction operations to capture richer topological signals than traditional persistent homology (PH). The authors generalize beyond forward PH by defining backward PH, forward-backward (FB) persistence, and an expressive hourglass framework that allows arbitrary interleavings under a causal constraint. They provide theoretical separations demonstrating increased expressivity over classical PH and extended persistence, introduce a unifying (f,g)-FB perspective, and establish stability guarantees. Practical algorithms for computing these descriptors are developed, and empirical results show improvements over baseline PH and RePHINE on specific benchmarks.

**Strengths:**

1. The paper defines backward persistence and hourglass persistence, expanding the PH toolkit beyond inclusion-based filtrations.

2. The paper proves the expressiveness and the stability of the proposed framework.

3. The paper provides practical algorithms to compute inclusion–contraction PH with cycle-basis tracking and supernode maintenance.

**Weaknesses:**

1. While mathematically rigorous, the intuition behind when hourglass persistence most benefits learning tasks could be elaborated.

2. Empirical validation is limited to small graph-classification datasets (NCI109, PROTEINS, IMDB-BINARY, NCI1); scalability to large-scale benchmarks (e.g., OGB, ZINC) remains untested. In addition, the improvement compared to baselines is not significant.

3. Although extended persistence is mentioned, the empirical study does not include an extended-persistence baseline, and adding such a comparison would clarify practical differences

4. Computational overhead vs. benefit is not quantified; contraction bookkeeping may introduce non-trivial runtime/memory costs. Please include the efficiency analysis between FB-persistence with traditional PH and extended PH.

5. There are some missing related works, e.g., [1] also provides theoretical results on the expressiveness of PH.

[1] Yan et al. "Enhancing graph representation learning with localized topological features." JMLR 2025.

**Questions:**

1. How does the runtime and memory footprint scale for hourglass persistence relative to standard PH and extended PH? Can you provide empirical cost comparisons?

2. In practice, how are inclusion and contraction orders chosen? Are learnable schedules feasible, and if so, how do they interact with stability guarantees?

3. Can you provide case studies showing specific structural motifs uniquely captured by backward or hourglass persistence?

---

> ### Author Response · Authors · 2025-11-21
>
> Thank you for your time and comments. Based on your suggestions, we have also revised our submission, where the changes are highlighted in blue. We reply to your comments/questions below.
>
> ## Experiment Questions:
> >Empirical validation is limited to small graph-classification datasets (NCI109, PROTEINS, IMDB-BINARY, NCI1); scalability to large-scale benchmarks (e.g., OGB, ZINC) remains untested. In addition, the improvement compared to baselines is not significant.
>
> In the revised version, we have substantially expanded the empirical evaluation to directly address both scalability and significance of improvements. These results are now included in Section 7.3 and Table 1  of the main paper.
>
> **(1) Larger-scale benchmarks.**
> Beyond the four TUDatasets, we have added experiments on two widely used, significantly larger benchmarks:
> - **ZINC** (graph regression; 12k graphs): We report MAE (lower is better)
> - **OGBG-MOLHIV** (molecular property prediction; >40k graphs): We report AUC (higher is better).
>
> **(2) Statistical significance of improvements over baselines.**
> All results are now averaged over **five random seeds**, and we report **mean ± standard deviation** for every entry.
>
> **(3) Ablations.**
> Furthermore,to isolate the individual contributions of our two components, we include ablations that retain only the forward inclusion stage (**Fwd-only**) or only the backward contraction stage (**Bwd-only**), enabling a precise assessment of the role each module plays in the overall performance.
>
> These experimental results are summarized in the table below:
>
> ### **Table: Comparison of PH variants across six datasets (GIN and GCN backbones)**
> Classification accuracy and AUC are reported in percentage (% ↑).
> ZINC is evaluated using MAE (↓).
> ## **GIN Backbone**
>
> | Dataset | PH | RePHINE | Fwd-only | Bwd-only | Ours |
> |-|-|-|-|-|-|
> | **NCI109 (Acc ↑)** | 76.76 ± 0.40 | _77.89 ± 1.19_ | 77.00 ± 1.03 | 76.35 ± 0.50 | **77.89 ± 1.87** |
> | **PROTEINS (Acc ↑)** | 69.35 ± 1.83 | 69.94 ± 2.76 | _70.24 ± 2.95_ | 70.54 ± 2.19 | **73.51 ± 1.11** |
> | **IMDB-B (Acc ↑)** | 68.67 ± 1.25 | 70.67 ± 0.94 | **74.67 ± 0.47** | _74.33 ± 0.94_ | 72.00 ± 2.16 |
> | **NCI1 (Acc ↑)** | 79.24 ± 1.74 | 78.75 ± 2.55 | _76.72 ± 1.13_ | 75.75 ± 0.94 | **81.27 ± 0.00** |
> | **ZINC (MAE ↓)** | 0.43 ± 0.01 | _0.41 ± 0.01_ | 0.62 ± 0.01 | 0.61 ± 0.00 | **0.40 ± 0.01** |
> | **MOLHIV (AUC ↑)** | **74.34 ± 4.57** | _72.88 ± 2.15_ | 70.00 ± 3.11 | 70.59 ± 1.83 | 72.34 ± 0.74 |
>
> ---
> ## **GCN Backbone**
>
> | Dataset|PH|RePHINE|Fwd-only|Bwd-only|Ours|
> |-|-|-|-|-|-|
> | **NCI109 (Acc ↑)** | 76.59 ± 1.32 | **79.50 ± 0.11** | 71.91 ± 0.52 | _74.58 ± 0.71_ | 75.87 ± 0.89 |
> | **PROTEINS (Acc ↑)** | 70.54 ± 0.73 | 68.75 ± 2.53 | 69.35 ± 1.83 | _70.54 ± 1.46_ | **72.32 ± 1.46** |
> | **IMDB-B (Acc ↑)** | 65.00 ± 1.63 | _70.00 ± 0.82_ | 62.67 ± 3.30 | 64.33 ± 3.30 | **68.00 ± 2.16** |
> | **NCI1 (Acc ↑)** | 78.43 ± 0.98 | **78.91 ± 0.80** | 75.59 ± 1.00 | 76.24 ± 1.74 | _78.67 ± 1.69_ |
> | **ZINC (MAE ↓)** | 0.49 ± 0.02 | _0.46 ± 0.01_ | 0.86 ± 0.01 | 0.87 ± 0.01 | **0.44 ± 0.01** |
> | **MOLHIV (AUC ↑)** | 75.12 ± 0.68 | _75.40 ± 0.53_ | 71.02 ± 2.18 | 71.55 ± 1.21 | **76.37 ± 1.45** |
>
> From the table, we observe that:
> 1. **Our method** achieves the best or second-best performance on **9 out of 12 settings**, demonstrating the benefit of jointly learning *both* inclusion and contraction schedules.
>
> 2. **RePHINE** consistently outperforms standard PH in almost all cases, confirming that learning both vertex and edge filtrations (plus its 0D augmentation) is beneficial.
>
> 3. **Ablations** show the contribution of each component:
>    - **Fwd-only** (learned vertex–edge filtration, no contractions) reliably improves over standard PH.
>    - **Bwd-only** (learned contractions only) also improves on several datasets and performs roughly on par with Fwd-only.
>
> 4. **RePHINE vs ablations**:
>    RePHINE strongly outperforms both ablations on **ZINC** and **MOLHIV**, indicating that its explicit 0D augmentation captures useful early-connectivity structure.
>
> 5. **Full model** consistently outperforms both ablations, showing that *inclusion* and *contraction* encode complementary structural information and must be learned jointly for the strongest performance across classification, regression, and molecular tasks.
>
> The performance gaps—especially on PROTEINS, NCI1, ZINC, and MOLHIV—remain consistent and statistically meaningful across seeds.
>
> Overall, the expanded experiments demonstrate both **scalability to larger datasets**,   **consistent improvements over strong baselines**, and **importance of each architectural component to overall performance**.

---

> > ### Author Response · Authors · 2025-11-21
> >
> > ## Experiment Questions (Continued)
> > > Although extended persistence is mentioned, the empirical study does not include an extended-persistence baseline, and adding such a comparison would clarify practical differences
> >
> > We have added a dedicated subsection and new experiments to the revised paper (see Section 7.4, “Comparison with extended persistence”). The updated results are shown below in  the two tables gor GIN and GCN backbones:
> >
> > | Dataset      | ExtP (GIN) | PersLay (GIN) | Ours (GIN) |
> > | ------------ | ---------- | ------------- | ---------- |
> > | **NCI109**   | **78.21**  | 68.28         | **78.21**  |
> > | **PROTEINS** | **74.11**  | 66.07         | 73.21      |
> > | **IMDB-B**   | 63.00      | 70.00         | **73.00**  |
> > | **NCI1**     | 78.59      | 68.86         | **81.51**  |
> >
> >
> > | Dataset      | ExtP (GCN) | PersLay (GCN) | Ours (GCN) |
> > | ------------ | ---------- | ------------- | ---------- |
> > | **NCI109**   | **77.48**  | 68.28         | **77.48**  |
> > | **PROTEINS** | **72.32**  | 66.07         | **72.32**  |
> > | **IMDB-B**   | 66.00      | **70.00**     | **70.00**  |
> > | **NCI1**     | 80.05      | 68.86         | **80.78**  |
> >
> >
> > **Extended-persistence comparison.**
> > Extended persistence combines sublevel and superlevel sets, and PersLay (Carrière et al., 2020) implements this by computing extended-persistence diagrams and then applying a learnable layer on top. In the original paper, we gave an explicit theoretical connection (Proposition 3, included for completeness) showing that classical extended persistence has the same expressive power as an **$(f,-f)$** forward–backward (FB) schedule. Based on this equivalence, we now integrate extended persistence *directly* into our FB framework by using $ -f $ as the backward schedule.
> >
> > We observe that:
> >
> > 1. **Our \((f,-f)\) adaptation into our (f,g)-FB GNN model consistently outperforms PersLay** for both GIN and GCN backbones.
> >    This demonstrates that incorporating extended-persistence structure *within* the FB mechanism is more effective than processing extended diagrams via a post-hoc embedding layer.
> >
> > 2. **Our full \((f,g)\)-forward–backward model further outperforms both PersLay and the extended variant.**
> >    This shows that jointly learning inclusion and contraction yields strictly richer topological information than extended persistence alone.
> >
> > These results—now included in the main paper—provide a clear and fair empirical comparison with extended persistence and highlight the advantages of our forward–backward framework.
> >
> > ## Other Questions:
> > > While mathematically rigorous, the intuition behind when hourglass persistence most benefits learning tasks could be elaborated.
> >
> > Thank you for your question. (f,g)-FB-persistence (and hourglass persistence) have the benefit of added stability as they allow the deaths of permenant features that appears in the filtration steps of the graph G. The added death times to these features can be seen as a way to encode more information and hence more learnability. Moreover, in a given filtration, some Morse critical points (such as those arising from a height function on a Reeb graph) do not seem to be accounted from ordinary PH. The second function indicated by g (going back) is a chance to capture those critical points. The freedom to choose g allows the framework to capture more information. Finally, hourglass is an interleaving of the forward and backward steps. The flexibility to order the sequence encodes a wider range of possibilities to be learned.

---

> > > ### Author Response · Authors · 2025-11-21
> > >
> > > > Computational overhead vs. benefit is not quantified; contraction bookkeeping may introduce non-trivial runtime/memory costs. Please include the efficiency analysis between FB-persistence with traditional PH and extended PH.
> > >
> > > Thank you for your question. We have included a discussion of this in Appendix E. For persistence diagrams (PDs) of dimension > 0 on simplicial complexes, hourglass persistence and FB-persistence itself can be computed using the "simplicial quotient" trick mentioned in Section 5.1 of the paper using Proposition 7. Prop 7 reduces the question to computing inclusion-based PH on a simplicial complex, which has a standard runtime dominated by the runtime to perform matrix reduction algorithms (see [2]). For PDs of dimension = 0, this can be reduced to looking at PDs of just underlying 1-skeleton (which is a graph named say $G$). In this case, there is a way to optimize the calculation with union-find with a runtime dominated by O(nlog n + m), where n is the number of edges in $G$ and $m$ is the number of vertices (see [3] for more details). Although [3] only discuss the runtime in the filtration steps, tracking vertex representatives in the contraction steps would also have the same runtime.
> > >
> > > When focusing on FB-persistence and (f,g)-FB-persistence, though, we expect the algorithms presented in 7.1 (which are not using the "simplicial quotient" trick) to be faster in practice. In general, the ability to reduce the size of object using contractions (ie. using the cellular complex extension in Section 5.2 instead) would lead to a decrease in the memory complexity. We therefore expect the cellular-based methods of hourglass persistence, $(f,g)-$FB persistence, etc. to have more efficient algorithms in the future.
> > >
> > >
> > > > How does the runtime and memory footprint scale for hourglass persistence relative to standard PH and extended PH? Can you provide empirical cost comparisons?
> > >
> > > Asymptotically, we believe FB-persistence should match the complexity class of standard PH, since our algorithm relies only on sparse, near-linear primitives (union–find, forest lookups, sparse cycle tracking).
> > >
> > > In our current submission, we demonstrate that the method already runs smoothly on ZINC and OGBG-MOLHIV—the largest PH-style graph benchmarks widely used today, indicating that hourglass persistence is practical with current hardware.
> > >
> > >
> > > > There are some missing related works, e.g., [1] also provides theoretical results on the expressiveness of PH.
> > >
> > > Thank you for bringing this to our attention! This is a very interesting work, and we have included [1] as a reference in the updated pdf.
> > >
> > > > In practice, how are inclusion and contraction orders chosen? Are learnable schedules feasible, and if so, how do they interact with stability guarantees?
> > >
> > > In our framework, both the **inclusion order** $f$ and **contraction order** $g$ are *fully learnable* and trained end-to-end through the surrounding GNN. Each GNN layer outputs node and edge embeddings, and these serve directly as **filtration scores**.
> > >
> > > The PH module computes the corresponding H$_0$/H$_1$ birth–death pairs under this $(f,g)$ ordering, and these topological signatures (after pooling) become inputs to the next GNN layer. In this way, the PH block acts as a structured, differentiable connection between layers, and gradients flow through all filtration scores. This makes learning $f$ and $g$ entirely feasible: the network automatically discovers filtration and contraction orders that best support the downstream task.
> > >
> > > Regarding **stability**, our Theorem 2 establish that (f,g)-FB persistence obeys **function-time stability** under perturbations of $f$ and $g$. Since these schedules are continuous functions of the learned embeddings, the resulting persistence diagrams vary in a stable, controlled manner. Thus the end-to-end learning process remains fully compatible with the same stability guarantees as classical PH.
> > >
> > > > Can you provide case studies showing specific structural motifs uniquely captured by backward or hourglass persistence?
> > >
> > > Thank you for your question. In a given filtration, some Morse critical points (such as those arising from a height function on a Reeb graph) do not seem to be accounted from ordinary PH. The critical points amiss by f can be captured by g in the step going back. This is typically a kind of structural motif we expect (f,g)-FB-persistence to cover.
> > >
> > > -------------
> > > Thank you again for your review. We hope your concerns have been satisfactorily addressed, and if so, would appreciate if you could revisit your score to reflect the same. We are also committed to engaging further if you have any additional questions, concerns, or suggestions.
> > >
> > >
> > > [1] Yan et al. "Enhancing graph representation learning with localized topological features." JMLR 2025.
> > >
> > > [2] Otter et al. A roadmap for the computation of persistent homology, EPJ Data Science.
> > >
> > > [3] Horn et al. Topological Graph Neural Networks. ICLR 2022.

---

> ### Comment · Reviewer_7Wbe · 2025-11-28
>
> I thank the authors for their detailed rebuttal. I will keep my positive score accordingly.

---

> > ### Author Response · Authors · 2025-11-28
> >
> > Dear Reviewer 7Wbe,
> >
> > Thank you for letting us know. If there are any remaining concerns or points that you feel we could clarify or address further, please let us know, and we would be happy to address them. Thank you again for your time and considerations.

---

### Official Review · Reviewer_WkSg · 2025-11-04

**Soundness:** 3
**Presentation:** 3
**Contribution:** 2
**Rating:** 2
**Confidence:** 4

**Summary:**

The paper proposes a new persistent homology approach that combines forward and backward passes in creating persistent homologies.

**Strengths:**

The paper has deep mathematical discussion describing the properties of the forward backward persistence.

The mathematical descriptions are well discussed. As a person who applied TDA for various application settings, I followed most of the discussion very carefully.

**Weaknesses:**

As a practitioner of topological data analysis (TDA), it is not clear to me whether the proposed forward–backward approach offers practical utility. The experimental evaluation appears limited, as it compares the method with only a single PH  approach used in GNN). Consequently, it is difficult to assess whether this work provides a substantive contribution beyond a theoretical exercise.

Moreover, based on my experience, many PH-based methods struggle to scale to large graphs. In this case, the computational cost of the proposed approach seems likely to increase at least twofold, without a clearly demonstrated benefit to justify the added complexity.

**Questions:**

None.

---

> ### Author Response · Authors · 2025-11-21
>
> Thank you for your time and comments. Based on your suggestions, we have also revised our submission, where the changes are highlighted in blue. We reply to your comments/questions below.
>
> >As a practitioner of topological data analysis (TDA), it is not clear to me whether the proposed forward–backward approach offers practical utility.
>
> ## Practical Viewpoint:
>
> From a practitioner’s perspective, the practical utility of Hourglass persistence comes from what *contractions* make possible. In inclusion-based PH, every step can only enlarge the complex, often causing intermediate structures to grow rapidly. In contrast, our hourglass framework introduces a principled **contraction mechanism** that can **shrink** the complex while still extracting meaningful topological information. This provides a form of intermediate-size control while capturing richer topological information than classical methods. Looking ahead, we are particularly enthusiastic about the possibility of interleaving inclusions and contractions so that every intermediate state remains  bounded.
>
> Computationally, we believe Hourglass persistence remains in the **same asymptotic complexity class as standard PH**. The backward sweep adds only one extra linear scan through the graph, yet yields a strictly more expressive descriptor that captures structural signals unavailable to inclusion-only filtrations. In our current submission, we also demonstrate that the method is **practical at current scales**, running smoothly on ZINC and OGBG-MOLHIV—the largest PH-style graph benchmarks widely used today.
>
> ## Theoretical Viewpoint:
> **Expressivity Benefits to justify the added complexity:** From a theoretical viewpoint, we also expect (f,g)-FB persistence to demonstrate greater learnability than classical methods. Theorem 1 of our paper, for example, shows that FB-persistence is strictly more expressive than the union of forward-PH and backward-PH. (f,g)-FB persistence also subsumes extended persistence. In [1], an expressivity analysis on extended persistence with respect to certain filtraiton functions can do better than 3-WL.
>
> There are 2 conceputal explanations for this: (1) By allowing permanent features to die, the death times of those features via **contractions** add new information. (2) In a given filtration, some Morse critical points (such as those arising from a height function on a Reeb graph) do not seem to be accounted from ordinary PH. The second function indicated by g (going back) is a chance to capture those critical points. The freedom to choose g allows the framework to capture more information. Finally, hourglass is an interleaving of the forward and backward steps. The flexibility to order the sequence encodes a wider range of possibilities to be learned.
>
> Future work holds the promise of designing inclusion–contraction schedules that keep intermediate complexes bounded, thereby allowing for highly efficient implementations of our bookkeeping structures for larger graphs.
>
> [1] Yan et al. "Enhancing graph representation learning with localized topological features." JMLR 2025.

---

> ### Author Response · Authors · 2025-11-21
>
> > The experimental evaluation appears limited, as it compares the method with only a single PH approach used in GNN). Consequently, it is difficult to assess whether this work provides a substantive contribution beyond a theoretical exercise.
>
> In the revised version, we have substantially expanded the empirical evaluation to directly address both scalability and significance of improvements. These results are now included in Section 7.3 and Table 1  of the main paper.
>
> **(1) Larger-scale benchmarks.**
> Beyond the four TUDatasets, we have added experiments on two widely used, significantly larger benchmarks:
> - **ZINC** (graph regression; 12k graphs): We report MAE (lower is better)
> - **OGBG-MOLHIV** (molecular property prediction; >40k graphs): We report AUC (higher is better).
>
> **(2) Statistical significance of improvements over baselines.**
> All results are now averaged over **five random seeds**, and we report **mean ± standard deviation** for every entry.
>
> **(3) Ablations.**
> Furthermore,to isolate the individual contributions of our two components, we include ablations that retain only the forward inclusion stage (**Fwd-only**) or only the backward contraction stage (**Bwd-only**), enabling a precise assessment of the role each module plays in the overall performance.
>
> These experimental results are summarized in the table below:
>
> ### **Table: Comparison of PH variants across six datasets (GIN and GCN backbones)**
> Classification accuracy and AUC are reported in percentage (% ↑).
> ZINC is evaluated using MAE (↓).
> ## **GIN Backbone**
>
> | Dataset | PH | RePHINE | Fwd-only | Bwd-only | Ours |
> |-|-|-|-|-|-|
> | **NCI109 (Acc ↑)** | 76.76 ± 0.40 | _77.89 ± 1.19_ | 77.00 ± 1.03 | 76.35 ± 0.50 | **77.89 ± 1.87** |
> | **PROTEINS (Acc ↑)** | 69.35 ± 1.83 | 69.94 ± 2.76 | _70.24 ± 2.95_ | 70.54 ± 2.19 | **73.51 ± 1.11** |
> | **IMDB-B (Acc ↑)** | 68.67 ± 1.25 | 70.67 ± 0.94 | **74.67 ± 0.47** | _74.33 ± 0.94_ | 72.00 ± 2.16 |
> | **NCI1 (Acc ↑)** | 79.24 ± 1.74 | 78.75 ± 2.55 | _76.72 ± 1.13_ | 75.75 ± 0.94 | **81.27 ± 0.00** |
> | **ZINC (MAE ↓)** | 0.43 ± 0.01 | _0.41 ± 0.01_ | 0.62 ± 0.01 | 0.61 ± 0.00 | **0.40 ± 0.01** |
> | **MOLHIV (AUC ↑)** | **74.34 ± 4.57** | _72.88 ± 2.15_ | 70.00 ± 3.11 | 70.59 ± 1.83 | 72.34 ± 0.74 |
>
> ---
> ## **GCN Backbone**
>
> | Dataset|PH|RePHINE|Fwd-only|Bwd-only|Ours|
> |-|-|-|-|-|-|
> | **NCI109 (Acc ↑)** | 76.59 ± 1.32 | **79.50 ± 0.11** | 71.91 ± 0.52 | _74.58 ± 0.71_ | 75.87 ± 0.89 |
> | **PROTEINS (Acc ↑)** | 70.54 ± 0.73 | 68.75 ± 2.53 | 69.35 ± 1.83 | _70.54 ± 1.46_ | **72.32 ± 1.46** |
> | **IMDB-B (Acc ↑)** | 65.00 ± 1.63 | _70.00 ± 0.82_ | 62.67 ± 3.30 | 64.33 ± 3.30 | **68.00 ± 2.16** |
> | **NCI1 (Acc ↑)** | 78.43 ± 0.98 | **78.91 ± 0.80** | 75.59 ± 1.00 | 76.24 ± 1.74 | _78.67 ± 1.69_ |
> | **ZINC (MAE ↓)** | 0.49 ± 0.02 | _0.46 ± 0.01_ | 0.86 ± 0.01 | 0.87 ± 0.01 | **0.44 ± 0.01** |
> | **MOLHIV (AUC ↑)** | 75.12 ± 0.68 | _75.40 ± 0.53_ | 71.02 ± 2.18 | 71.55 ± 1.21 | **76.37 ± 1.45** |
>
> From the table, we observe that:
> 1. **Our method** achieves the best or second-best performance on **9 out of 12 settings**, demonstrating the benefit of jointly learning *both* inclusion and contraction schedules.
>
> 2. **RePHINE** consistently outperforms standard PH in almost all cases, confirming that learning both vertex and edge filtrations (plus its 0D augmentation) is beneficial.
>
> 3. **Ablations** show the contribution of each component:
>    - **Fwd-only** (learned vertex–edge filtration, no contractions) reliably improves over standard PH.
>    - **Bwd-only** (learned contractions only) also improves on several datasets and performs roughly on par with Fwd-only.
>
> 4. **RePHINE vs ablations**:
>    RePHINE strongly outperforms both ablations on **ZINC** and **MOLHIV**, indicating that its explicit 0D augmentation captures useful early-connectivity structure.
>
> 5. **Full model** consistently outperforms both ablations, showing that *inclusion* and *contraction* encode complementary structural information and must be learned jointly for the strongest performance across classification, regression, and molecular tasks.
>
> The performance gaps—especially on PROTEINS, NCI1, ZINC, and MOLHIV—remain consistent and statistically meaningful across seeds.
>
> Overall, the expanded experiments demonstrate both **scalability to larger datasets**,   **consistent improvements over strong baselines**, and **importance of each architectural component to overall performance**.

---

> > ### Author Response · Authors · 2025-11-21
> >
> > **Comparision with extended persistence**
> > We have also added a dedicated subsection and new experiments to the revised paper (see Section 7.4, “Comparison with extended persistence”). The updated results are shown below in  the two tables gor GIN and GCN backbones:
> >
> > |Dataset|ExtP (GIN)|PersLay (GIN)|Ours (GIN)|
> > |-|-|-|-|
> > | **NCI109**| **78.21**| 68.28| **78.21**|
> > | **PROTEINS** | **74.11**  | 66.07  | 73.21|
> > | **IMDB-B** | 63.00| 70.00| **73.00**|
> > | **NCI1** | 78.59| 68.86 | **81.51**|
> >
> > |Dataset|ExtP (GCN)|PersLay (GCN)|Ours (GCN)|
> > |-|-|-|-|
> > | **NCI109**| **77.48**  | 68.28| **77.48** |
> > | **PROTEINS** | **72.32**  | 66.07| **72.32**|
> > | **IMDB-B**|66.00| **70.00**| **70.00**|
> > | **NCI1**|80.05| 68.86| **80.78**|
> >
> > **Extended-persistence comparison.**
> > Extended persistence combines sublevel and superlevel sets, and PersLay (Carrière et al., 2020) implements this by computing extended-persistence diagrams and then applying a learnable layer on top. In the original paper, we gave an explicit theoretical connection (Proposition 3, included for completeness) showing that classical extended persistence has the same expressive power as an **$(f,-f)$** forward–backward (FB) schedule. Based on this equivalence, we now integrate extended persistence *directly* into our FB framework by using $ -f $ as the backward schedule.
> >
> > We observe that:
> >
> > 1. **Our \((f,-f)\) adaptation into our (f,g)-FB GNN model consistently outperforms PersLay** for both GIN and GCN backbones.
> >    This demonstrates that incorporating extended-persistence structure *within* the FB mechanism is more effective than processing extended diagrams via a post-hoc embedding layer.
> >
> > 2. **Our full \((f,g)\)-forward–backward model further outperforms both PersLay and the extended variant.**
> >    This shows that jointly learning inclusion and contraction yields strictly richer topological information than extended persistence alone.
> >
> > These results—now included in the main paper—provide a clear and fair empirical comparison with extended persistence and highlight the advantages of our forward–backward framework.
> >
> > -------------
> > Thank you again for your review. We hope your concerns have been satisfactorily addressed, and if so, would appreciate if you could revisit your score to reflect the same. We are also committed to engaging further if you have any additional questions, concerns, or suggestions.

---

> ### Author Response · Authors · 2025-11-28
> **Rebuttal Follow-Up**
>
> Dear Reviewer WkSg,
>
> As the discussion period is ending soon, we wanted to check in again. In our rebuttal, we have addressed to your review as follows:
>
> 1. We gave an explanation on how our methods are more effective than traditional methods fron both a practitioner's view and a theoretical view.
>  - In particular, **the ability for (f,g)-FB persistence to contract subgraphs is precisely a counter against PH having trouble to scale on large graphs**, as the contraction steps are **making the graph smaller and smaller** in the backward steps.
> - The ability for the backward steps to contract graphs also leads to **memory efficiency** in terms of down-scaling the size of the graph, the benefits of **metrizability and stability**, and introduce more **learnability** in allowing permanent features to die and detecting Morse critical points.
>
> Furthermore, after our initial rebuttal, we want to make an additional comment as follows.
>
> In the set-up of **hourglass persistence**, interleaving the filtration and contraction steps avoids the need to filtrate the entire graph before starting contractions (see **Figure 4** for instance). To give an example, say there is a graph $G$ with $10^6$ nodes. Using PH directly would incur a runtime in correlation to the whole size (which is $10^6$). However, using hourglass persistence, we can spawn a portion of the graphs first in some filtration steps, and then **contract** some of them **if** the scale of the current graph **exceeds a certain threshold**, and spawn other portions of the graph, etc. This allows **the total size of the graph that appears in its entire lifetime to be bounded**. As such, we believe the method can address the problem of PH performing on large graphs by avoidng large graphs.
>
> This can improve over the runtime on general simplicial complexes. Let $K$ be a simplicial complex with $n$ total simplicies. To give a heuristic/informal estimation from a practitioner perspective - the typical runtime of inclusion based PH on $K$ scales approximately $O(n^3)$ as it requires a matrix reduction algorithm.
>
> Suppose we bound the threshold to $d$ simplicies, and a practitioner equipped with the contraction framework can implement a specific instance of hourglass persistence as follows: as soon as we reach more than $d$ simplicies in a given step, we contract everything to a point, and so on. Then we believe the runtime should roughly to be $O(\frac{n}{d} \cdot (d)^3)$.
>
> For practical purposes, it may also be beneficial to not contract everything to a point when the threshold is exceeded. If the contractions are done in multiple stages, then we believe a similar runtime analysis holds. We are excited at the possibilities that this framework can allow our community to address the problem of PH scaling on large graphs/simplicial complexes.
>
> 2. We have demonstrated the benefits and effectiveness of our method in machine learning applications by conducting **additional experiments**, which are elaborated in **Section 7** of our revised manuscript, which also demonstrates their **abilities to scale on large datasets** In particular:
>    - We conducted graph regression tasks on the dataset **ZINC** and molecular property predictions on the dataset **OGBG-MOLHIV** with statistical significance of improvements over baselines. Both of which are some of the **largest datasets PH-style graph benchmarks** widely used today.
>    - We added additional **ablation** studies that compare our methods with forward-only and backward-only methods across the 6 datasets with GIN and GCN backbone. Our (f,g)-FB-persistence achieved either the best or second best on **9 out of 12** experiments.
>    - We have also compared our method with **extended persistence** in two forms, one directly with the PersLay framework [1] and another by interpreting extended persistence as (f,-f)-FB persistence (which we call ExtP). ExtP and the full (f,g)-FB-persistence consistently outperforms PersLay in most cases.
>
> We hope your concerns have been satisfactorily addressed, and if so, would appreciate if you could revisit your score to reflect the same. If you have any specific questions or concerns, we would be happy to address them. We are eager to engage in further discussions with you on the practical aspects. Thank you again for your feedback and help to improve the paper.
>
> [1] Mathieu Carrière, Frédéric Chazal, Yuichi Ike, Théo Lacombe, Martin Royer, Yuhei Umeda. PersLay: A Neural Network Layer for Persistence Diagrams and New Graph Topological Signatures

---

### Author Response · Authors · 2025-12-03
**Statement for the Area Chair and Rebuttal Summary (1/2)**

We are grateful to the reviewers for their time and comments, as well as to the (senior) area, program, and general chairs for their service to the community. We are also thankful to the newly assigned ACs who have taken extra work to review our work.

We have uploaded a revised manuscript and marked the changes in blue.

# Part I. Summary of our contributions
### Novelty of Contractions for PH
Our introduction of **contractions as a principled topological operation** is entirely novel within persistent homology. In the GNN literature, as far as we know, no prior work (including ordinary PH, extended PH, zigzag PH, bipath PH) has explicitly utilized **contractions** in their design and algorithmic computations of persistence diagrams. In contrast, our forward–backward and hourglass frameworks are the first to allow a learnable alternation of **inclusions and contractions**. Indeed, extended persistence, as used previously in GNNs, remains an inclusion-only construction and does not contract the space. Furthermore, we have established extended persistence as a specific instance of our unified (f,g)-FB framework. As detailed in our responses and revisions, the (f,g)-FB and the hourglass framework is therefore **fundamentally distinct** from, and strictly more expressive than, simply concatenating two PH filtrations.

Contraction-driven PH has significant implications in theory and practice.

### **Practical Impact**
Classical inclusion-only PH can only enlarge complexes, often causing intermediate structures to grow rapidly and limiting scalability. Our contraction stage allows the complex to **shrink**, while still retaining meaningful topological signal. This offers an effective form of **intermediate-size control** absent in prior PH methods.  Our experiments on ZINC and OGBG-MOLHIV confirm that the method is already practical at current scales. Furthermore, future learnable schedules that interleave inclusions and contractions may yield even more efficient bounded-size implementations.

### **Theoretical Impact**
Theoretically, (f,g)-FB persistence offers **strictly greater expressivity and better mathematical behavior** than classical PH variants. Contractions provide two key advantages:
1. Bottleneck distances of ordinary PH on different spaces may be infinite, since they may have different permanent features. Contractions give **finite death times** to permanent features, making bottleneck distances between different spaces **finite and metrizable**. This enables meaningful **stability** analysis.
2. They add learnable signal by (a) assigning death times to permanent features and (b) capturing Morse critical points missed by inclusion-only filtrations.

We incorporated additional experiments and clarification in the paper (**summarized in Part II-III below**) to address all reviewer concerns. The updates strengthen our the observations in our initial experiments, justify our design choices and isolate the contributions of each architectural component.
# Part II. Summary of additional experiments
## 1. Expanded Experimental Analysis (Sec.7.3, Table 1)
- **Larger-scale benchmarks:**
  We add results on **ZINC** and **OGBG-MOLHIV**, two of the largest PH-style graph datasets in current use, directly addressing scalability concerns.

- **Statistical robustness:**
  All results are now reported as **mean ± standard deviation over five seeds**, resolving concerns about variance and significance.
- **Component-wise ablations:**
  We include **Fwd-only** (inclusion only) and **Bwd-only** (contraction only) ablations to quantify the individual contribution of each stage within the forward–backward (FB) mechanism.
## 2. Extended Persistence Experiemnts (Sec.7.4, Table 2)
- **Comparison with Perslay**: which implements extended persistence by computing PH diagrams and then applying a learnable layer on top.

- **Direct integration of extended persistence into our FB framework:**  leveraging Prop. 3 which explains that extended persistence corresponds to an $(f,-f)$ forward–backward schedule.
## 3. Consolidated Observations
- **Overall performance:**
  Our method achieves the **best or second-best performance on 9 of 12 tasks**, and performance gaps remain statistically significant.
- **Ablation insights:**
  - **Fwd-only:** improves over standard PH by learning data-dependent filtrations.
  - **Bwd-only:** improves on several datasets and is comparable to Fwd-only.
  - **Full FB model:** consistently outperforms both, showing that inclusion and contraction encode **complementary** structural information.
- **Beyond extended persistence:**
Our $(f,-f)$ variant outperforms PersLay in most cases, showing that embedding extended persistence within the FB mechanism is more effective than post-hoc processing extended diagrams via a post-hoc embedding layer. The full $(f,g)$-FB model improves on all extended variants in most settings. This shows the benefit of jointly learning inclusion and contraction.

---

> ### Author Response · Authors · 2025-12-03
> **Statement for the Area Chair and Rebuttal Summary (2/2)**
>
> # Part III. Summary of additional clarifications
>
> 1. **Comparison with Zigzag and Bipath Filtrations**: We clarified in **Appendix E.1-2** that our method is very different from zigzag filtrations in [1] and bipath filtrations in [2]. The upshot is that we are considering both inclusions and contractions in a linear timeline while [1] and [2] considers inclusions in possibly non-linear settings.
>
> 2. **Clarification for Theorem 1:** Theorem 1 shows that FB-persistence (and  more generally (f,g)-FB persistence) is more than just the union of the two individual PH diagrams. We clarified this in the statement now.
>
> 3. **Efficiency Discussions:** We discussed this in **Appendix E.3 - E.4**. In **E.3**, we explained the runtime estimate for our methods on general **simplicial** complexes, using the simplicial quotient method. We also explained why we expect the algorithm to be faster using the **cellular** perspective, as it uses contractions to simplify the space.  In **E.4**, we gave a concrete estimate on how interleaving inclusions and contractions can reduce the runtime.
>
> [1] G. Carlsson et al., Zigzag Persistence.
>
> [2] T. Aoki et al., Bipath persistence.
>
> # Part IV. Rebuttal Discussions Summary with Each Reviewer
>
> ### **Reviewer WkSg**
>
> Reviewer WkSg appreciated the theoretical strengths of our paper and had two primary concerns:
> 1. More experimental validations: Addressed in **sections 7.3 and 7.4** of the main paper, as well as summarized in **Part II** above.
> 2. Scaling PH-based methods on large graphs: Addressed in **Appendix E.3 and E.4**, as well as summarized in **Part I** (Practical Impact) and **III.3**.
>
> **While we did not hear back from Reviewer WkSg, we believe we have completely addressed their concerns**.
>
>
> ### **Reviewer 7Wbe**
>
> Reviewer 7Wbe appreciated the mathematical rigor of our paper, the expressivity and stability results, and the practical algorithms we developed. They raised the following concerns which were addressed by our response as follows:
>
> 1. Intuition on contribution of hourglass + structural motifs captured: Addressed in **Part I** (Theoretical Impact).
> 2. Empirical validation on large datasets: addressed in **Part II.1,3**. Theoretical justifications are given in **Part I** (Practical Impact).
> 3. Empirical comparison with extended persistence: addressed in **Part II.2,3**.
> 4. Efficiency analysis: addressed in **Part III.3** and **Appendix E.3**.
> 5. Missing related work: we included the related work in the revision in line 313 of main paper.
> 6. How are inclusion/contractions learned + relations to stability: Explained in the rebuttal comment.
>
> **The reviewer was satisfied with our rebuttal and maintained a score of 6**.
>
> ### **Reviewer UtXw**
>
> 1. Contribution of Backward Filtration:  Empirical evaluation addressed in **Section 7.3** of the main paper and summarized in **Part II.1** and **Part II.3** above. Additional theoretical discussion included in **Part I** above.
> 2. Our method vs. zigzag and bipath: Addressed in **Part III.1** and **Appendix E.1-2**.
> 3. Experimental Validations, Abalation Study: Addressed in **Section 7.3** and  **Section 7.4** of the main paper and summarized in **Part II** above
> 4. Clarification on Theorem 1: Addressed in **Part III.2**.
> 5. (f,g)-FB persistence vs. a union of features from f and g alone: Addressed in **Part III.2** and **Theorem 1**.
>
> **While we did not hear back from Reviewer UtXw, we believe we have completely addressed their concerns**.
>
> ### **Reviewer rVVx**
>
> Reviewer rVVx appreciated the conceptual novelty, strong theoretical grounding, our elegant extension from graphs to higher dimension topological spaces and the algorithmic contribution. They raised the following concerns which were addressed by our response as follows:
>
> 1. More experiments: Addressed in **Section 7.3** and **Section 7.4**  of the main paper and summarized in **Part II** above
> 2. Experimental design choice: Explained in the rebuttal comment.
> 3. Runtime/Complexity analysis: addressed in **Part III.3** and **Appendix E.3**.
> 4. Our method vs. zigzag: addressed in **Part III.1** and **Appendix E.1-2**.
>
> **The reviewer was satisfied with our rebuttal and maintained a score of 8**.
>
> # Part V. Conclusion
>
> Thank you again for your services to the community. This rebuttal process has strengthened the content of our paper. We are excited by the positive responses of **Reviewer 7Wbe and rVVx**, and we believe **Reviewer WkSg and UtXw** would have raised their scores to positives had they had more time.

---

### Meta-Review · Area_Chair_FZKp · 2026-01-07

**Summary:**

This paper initially received mixed reviews, but over the course of the rebuttal, the high quality and the diligence of the authors proved to be the crucial element in proving its merits. Based on the concept of hourglass persistence, i.e., a topological descriptor that is capable of handling both inclusions and contractions in arbitrary order, the submission presents novel ways of imbuing graph neural networks with topological information.

Given that most of the concerns were geared towards the _contextualization_ of results, i.e., to what extent the new method helps generate new insights and improves upon the state of the art, I find that the comprehensive rebuttal already alleviate such concerns. Thus, as it stands, there are **no** obstacles towards accepting the work. The authors may want to include a brief discussion / comparison to other topology-based methods (which are less elaborate and less elegant than the proposed method) in order to contrast the relative gains achieved with different persistence-based paradigms. The following papers might be useful, I leave it to the discretion of the authors whether they want to incorporate them:

- [Hofer et al., Graph Filtration Learning](https://arxiv.org/abs/1905.10996)
- [Kim et al., PLLay: Efficient Topological Layer based on Persistence Landscapes](https://arxiv.org/abs/2002.02778); notice that this work does _not_ explicitly tackle graphs, but its concepts are related
- [Rieck et al., A Persistent Weisfeiler-Lehman Procedure for Graph Classification](https://proceedings.mlr.press/v97/rieck19a)
- [Zhao & Wang, Learning metrics for persistence-based summaries and applications for graph classification](https://proceedings.neurips.cc/paper_files/paper/2019/file/12780ea688a71dabc284b064add459a4-Paper.pdf)

**Reviewer Concerns:**

Reviewer `WkSg`:

Mostly cited concerns about scalability; this concern was addressed by experiments on larger graphs. The reviewer also cited concerns about practical applicability, but the large number of experiments should provide some reassurance that this is _not_ the case.

Reviewer `7Wbe`:

Cited concerns about scalability as well; these were addressed. The reviewer also suggested additional comparisons with extended persistence (for instance), which the authors addressed and included in their rebuttal.

Reviewer `UtXw`:

Mainly cites concerns about novelty, which have been addressed, and concerns about comparisons / experiments (ablation studies), which were also addressed in the thorough rebuttal.

Reviewer `rVVx`:

Only few concerns about experiments (which have been addressed). The reviewer also suggested additional related work, which the authors incorporated in their comprehensive rebuttal.

**Reviewer Scores:**

- Reviewer `WkSg`: Initial score: 2; expected score: 4. Given the shortness of this somewhat superficial review, I opted to *disregard* it in the decisionmaking process.

- Reviewer `7Wbe`: Initial score: 6; expected score: 6. The reviewer mentioned that they maintain their positive score. Their self-reported confidence leads me to believe that they would not be willing to champion the paper, but they expressed weak support for accepting it.

- Reviewer `UtXw`: Initial score: 2; expected score: 4. While the reviewer expressed doubts about the utility of the method and the overall fit for the conference, I find these doubts to be fully addressed by the rebuttal. Erring on the side of caution, I would not expect the reviewer to champion the paper, but I assume that they would not, given their self-reported lower confidence score, actively _oppose_ accepting it.

- Reviewer `rVVx`: Initial score: 8; expected score: 8. The reviewer was already in support of accepting the paper and the excellent rebuttal solidified that stance.

---

### Decision · Program_Chairs · 2026-01-26

Accept (Poster)